# Ice-sheet hydro-fracture not advanced inland by lower-elevation lake drainages in Kalaallit Nunaat

Laura A. Stevens [1,2] ✉, Meredith Nettles [3,4], Stacy Larochelle [3,5], Marianne Okal [6], Emily Falconer[1], Natalie Turner[1], Joshua Rines [7], Ching-Yao Lai [7] & George Lu [3]

Drainage of supraglacial lakes via hydro-fracture is widely argued to be a mechanism for destabilization of grounded ice sheets under climate-warming scenarios because it may accelerate surface meltwater access to the ice-sheet bed. Progress in interrogating this hypothesis has been hindered by the lack of regional observations of hydro-fracture event occurrence, and the lack of observations of regional ice-sheet response to hydro-fracture events. Here, we remedy both deficiencies using a 22-station Global Navigation Satellite System array to discern inter-lake, hydro-fracture-event triggering potential between lakes spanning the mid-to-upper Greenland Ice Sheet ablation zone. In four separate instances, multiple lake hydro-fracture events occur close in time at similar elevations; meanwhile, strain rates across higher-elevation lake basins are unperturbed. Our findings support a simple model for the inland progression of surface-to-bed meltwater pathways beneath lakes: pathway initiation migrates alongside advancing surface melt, but is not accelerated by drainage activity at distant, lower-elevation lakes.

The hypothesis that supraglacial lake drainage via hydro-fracture[1] may migrate inland far faster than climate-driven melt has often been presented[2-7] as a mechanism for regional-scale destabilisation of grounded-ice-sheet flow during periods of climate warming. This hypothesis is attractive: hydro-fracture events generate the largest, short-timescale ice-sheet flow accelerations[1,8-11], and inter-lake triggering of lake drainage by hydro-fracture demonstrably occurs between neighbouring lakes[9,12]. A long-range-triggering hypothesis hinges, however, on applications of theoretical work on longitudinal stress coupling within the ice[13-15] to argue that surface-melt-driven ice-flow acceleration at lower elevations[16,17] produces ice-flow acceleration[14,18,19], and hydro-fracture beneath surface lakes[20], at higher elevations. An inland acceleration of surface-meltwater access to the ice-sheet bed beneath lakes[20] might then outpace the observed, climate-driven altitudinal advance of surface lakes[21].

Recently, *Christoffersen* et al.[20] have argued that satellite observations of lake-volume reduction[22] show >100 hydro-fracture events across tens-of-kilometre distances to occur within days; they conclude that these observations demonstrate long-range, hydro-fracture-event triggering as the mechanism for over three quarters of lake-drainage events, and as a mechanism for the rapid inland expansion of surface-to-bed meltwater pathways beneath lakes[20]. The lake-volume reduction events[22] that *Christoffersen* et al. analysed occur within a southwestern region of the Greenland Ice Sheet where ~ 90% of lakes transfer meltwater downstream from one lake to the next, or from one lake to a downstream moulin, through supraglacial river outlets that form whenever lakes overtop their basins[23,24]; and where an estimated

[1]Department of Earth Sciences, University of Oxford, Oxford, UK. [2]Radcliffe Institute for Advanced Study at Harvard University, Cambridge, MA, USA. [3]Lamont-Doherty Earth Observatory, Palisades, NY, USA. [4]Department of Earth and Environmental Sciences, Columbia University, New York City, NY, USA. [5]Department of Earth, Planetary and Space Sciences, University of California Los Angeles, Los Angeles, CA, USA. [6]EarthScope Consortium, Boulder, CO, USA. [7]Department of Geophysics, Stanford University, Stanford, CA, USA. ✉e-mail: laura.stevens@earth.ox.ac.uk

~ 72–86% of lakes lose volume gradually over multiple days[22,25]. However, the analysis of *Christoffersen* et al. does not differentiate between lake-volume reductions that transport meltwater laterally, from over-spilling lakes[22–25], and the comparatively infrequent, vertical transport of meltwater via hydro-fracture[1,8–12]. Instead, all lake-volume reduction events occurring at all surface lakes identified in the study region form the sample space for inferring temporal clusters of hydro-fracture events, yielding over four drainage events per lake, on average, during a single melt season[20]. Other studies show that hydro-fracture events rarely, if ever, occur more than once in a melt season at the same lake[1,8–12]. The data-analysis choices of *Christoffersen* et al. reflect common practice in remote observation of lake drainages in Kalaallit Nunaat[20,22,25–31], and the Antarctic[5,32], where temporal aliasing of satellite observations of lake-volume reduction blurs the mechanistic insight needed to identify where and when hydro-fracture events occur, and to determine the distances over which hydro-fracture-event triggering may act between lakes.

Here, we present the in-situ and remote observations needed to adequately test whether hydro-fracture events at lower-elevation lakes can trigger hydro-fracture events at higher-elevation lakes. With a 22-station Global Navigation Satellite System (GNSS) array installed around seven lake basins, we observe ice motion at adequate temporal sampling rates (15-s) to discern inter-lake, hydro-fracture-event triggering potential along a 55-km transect of the Greenland Ice Sheet ablation zone (Fig. 1). We capture eleven hydro-fracture events over our two-melt-season deployment, more than doubling global observations of this phenomenon[1,8–11]. We combine our field observations with a remote classification of lake-drainage mechanisms based on human identification of post-drainage, lake-basin features for the ~ 200 lakes that form annually near our GNSS array.

We identify temporal clusters of hydro-fracture events and interrogate them with several model-based approaches for bounding stress-interaction length scales and assessing physically plausible instances of inter-lake, hydro-fracture-event triggering. These instances amount to clusters of handfuls of lake-drainage events, whose small sizes comprise 2–7% of the lakes in the study region with volumes viable for full-ice-thickness hydro-fracture at the times that the clusters take place. Our ground-truth, GNSS observations repeatedly show no strain-rate perturbations above our detectable threshold ($\pm 0.002$ yr$^{-1}$) across higher-elevation ice-sheet regions and lake basins, while local clusters of hydro-fracture events transpire at lower elevation. This isolation between lower- and higher-elevation basins strongly argues that—although surface lakes advance higher in a warming climate[21]—the ability of this meltwater to access the bed and augment ice-flow speeds at high elevations is not accelerated by the process of long-range (i.e., over tens-of-kilometre distances), hydro-fracture-event triggering.

## Results and discussion
### Limitations of apparent-rapidity, lake-drainage classification
Our limited understanding of mechanisms underpinning the rate at which hydro-fracture events advance inland, and whether and how this inland expansion influences broad-scale ice-sheet dynamics[33], has motivated a decades-long effort to identify hydro-fracture events from satellite observations, despite their coarse temporal resolution (24–144 hr)[21,22,25–29,31,34,35]. The handful of hydro-fracture events observed by high-rate lake-level and GNSS recordings shows that lake drainage via hydro-fracture takes just 2–5 hr[1,8–11]. Immediately preceding hydro-fracture initiation, hours-long precursory periods of gradual lake-level lowering[1,8] and above-background rates of basal sliding[8,9] and ice-sheet uplift[8–10] occur. The finding that precursory ice deformation can facilitate stress accumulation within the lake basin of sufficient magnitude to promote ice fracture[9] has prompted a turn in the literature towards interpreting the timing between one hydro-fracture event and the next within a causal, triggering framework[12,16,17,20,31,35,36].

In making the causal interpretations that link drainage events, however, studies relying on remote-sensing analyses of lake drainage have routinely ignored multiple impacts of temporal aliasing[16,17,20,26,31]. Lakes may drain through moulins, by overspilling their banks, or by hydro-fracture. Most authors constructing regional lake-drainage catalogues partition "rapid" and "slow" drainages based on the time elapsed between pre- and post-drainage images[22,25–31], often using "rapid" classification as a proxy indicator for hydro-fracture[16,20,26,31]. However, author-chosen thresholds for "rapid" drainage are 48–144 hr[22,25–31], making the threshold for "rapid" drainage 10–72 times longer than a 2–5-hr hydro-fracture event[1,8–12]. This long window allows enough time within "rapid" time-elapsed thresholds for multiple hydro-fracture events to occur without overlapping in time, or for multiple drainage events to occur by lakes overspilling their banks[36,37]. Results from these apparent-rapidity, lake-drainage catalogues routinely include temporal clusters of "rapid" drainages[22,25,26,29,31,34,36], which are often interpreted as clusters of hydro-fracture events[16,17,20,26,31]. In addition, authors often assign a lake-drainage date corresponding to the first-available image of the drained basin[20,26,31], forcing drainage dates to align with cloud-free satellite images and thereby implicitly making cluster size a function of cloudiness and the temporal sampling interval[20,31].

The true picture of which lake-drainage events could plausibly exhibit inter-lake, hydro-fracture-event triggering[12] is also fundamentally obscured by glaciological and climatological conditions intrinsic to lake formation. Elevation-dependent runoff rates[38] cause nearby surface basins of similar sizes[39] to attain sufficient lake volumes for overspilling their basins[23,37] or driving water-filled fractures to the bed[40] at similar times[8,12,22,25,34]. Because lakes at similar elevations are on the same filling cycle, temporal clustering of lake drainage may arise stochastically. Studies to date incorporate neither elevation-dependent lake-filling nor lake-draining rates into their interpretations of clustered drainage events[16,17,20,26,31].

### Feature-based classification of lake-drainage mechanism
We remedy these deficiencies by conducting a feature-based classification of lake-drainage mechanisms to produce the first catalogue of drainage mechanisms, rather than apparent rapidity of drainage. Using the Fully Automated Supraglacial Lake Tracking at Enhanced Resolution (FASTER) algorithm[30], we track lake surface area for the ~ 200 lakes that surpass a minimum surface area of 0.165 km$^2$ and are located within a 7189-km$^2$, subglacial-catchment-delineated region of interest (ROI) encompassing our GNSS array (Fig. 1; Methods: Mechanistic lake-drainage catalogues). We visually assess drainage mechanisms through the inspection of lake-basin features, delineating four categories: drainage via hydro-fracture; drainage via a moulin within the lake margin; drainage via lateral overspill into a surface stream; and lakes that fill, but then freeze over, with no surface exits (Fig. 2).

Of the ~200 lakes that form within the ROI each melt season, our feature-based categorisation identifies 11–14% as draining via hydro-fracture (Fig. 3c, f), with bright, linear, kilometre-scale features that we interpret to be fracture scarps visible in immediate, post-drainage images (Fig. 2a). A further 11–14% of lakes deliver meltwater to the bed at the location of the lake basin via moulins that form within the maximum margin of the lake (Figs. 2b, 3c, f). Lateral movement of meltwater away from lake basins occurs for 63–69% of the lakes within the ROI, where we observe lakes to overspill their brims, but not to hydro-fracture or drain via an in-lake-margin moulin within the melt season (Figs. 2c, 3c, f). Roughly a quarter to a third of these overspilling lakes transfer meltwater into a downstream moulin, while the remaining majority of overspilling lakes transfer meltwater into another lake (Fig. 1b, c). We observe no-exit, frozen lakes predominantly in upper-elevation regions, with 6–12% of lakes observed to freeze over at the end of the melt season without having drained (Figs. 2d, 3c, f).

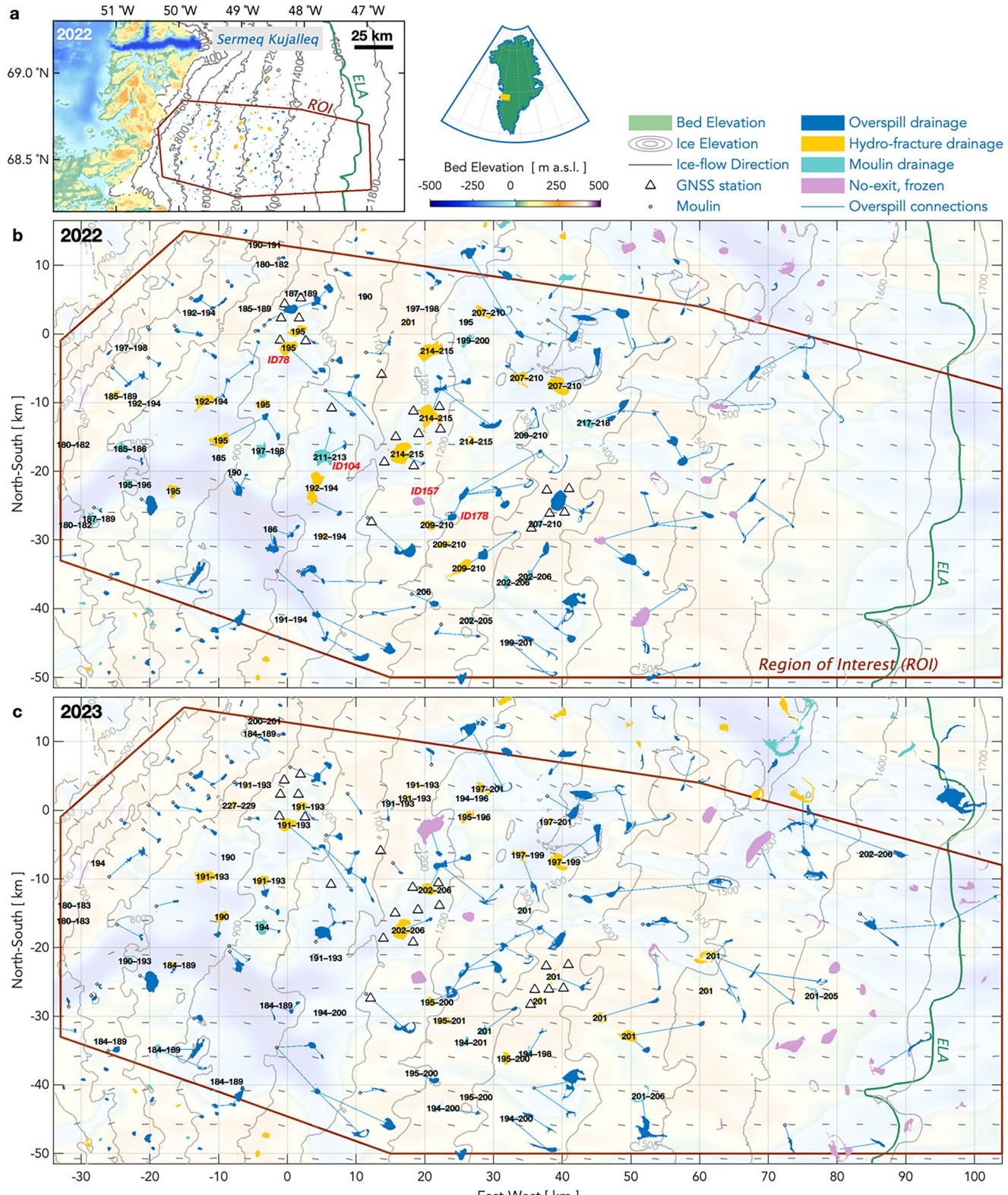

**Fig. 1 | Drainage mechanism for lakes in the region of the 2022 and 2023 GNSS arrays. a** Study region of interest (ROI) with lakes coloured by drainage mechanism in 2022 (see Legend). Green line shows long-term equilibrium line altitude (ELA) from 1958–2019[38]. **b, c** Grey triangles show the GNSS array in (**b**) 2022 and (**c**) 2023. Black numbers show days of year of hydro-fracture and moulin-drainage events identified in Sentinel-1/2 analysis. Red labels identify four lakes shown in Fig. 2. Blue lines show supraglacial connections between overspilling lakes; white circles show terminal moulins of overspilling lakes. Grey lines show 100-m ice-sheet surface elevation contours, and the colormap shows bed elevation from BedMachine v.5[63,64]. Grey ticks show ice-flow direction. Map origin is the 2011 position of the "North Lake" M1 moulin (68.72 °N 49.53 °W) identified in past work[9,12].

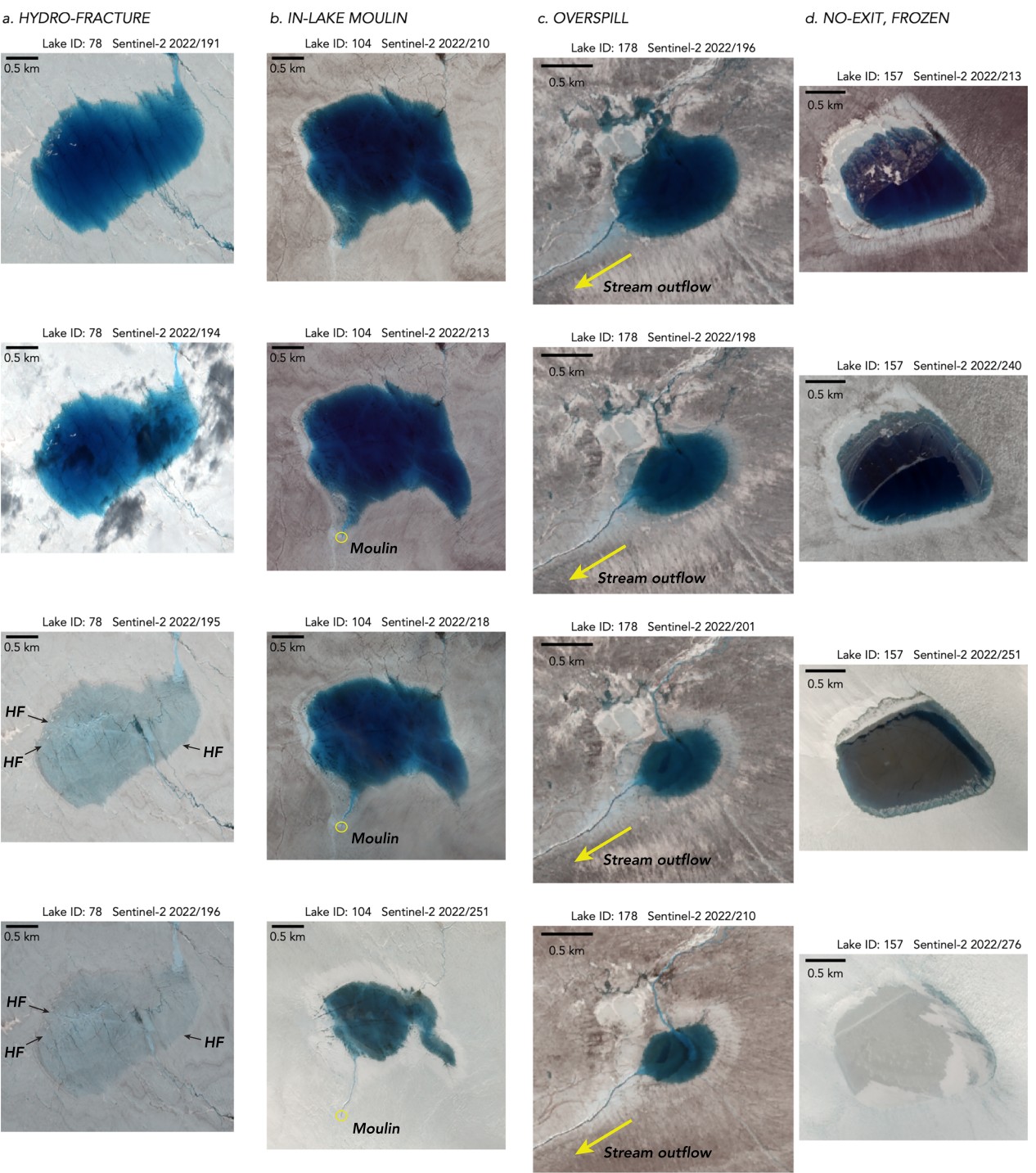

**Fig. 2 | Visual identification of drainage mechanism.** Select, Sentinel-2 image timeseries of four supraglacial lakes in 2022. Lake identification number and date of image provided above each panel; lake position shown in Fig. 1b with lake identification number. **a** Lake 2022-78 (68.70 °N 49.52 °W): drainage by hydro-fracture identified by the presence of linear, kilometre-scale features (black arrows) in immediate, post-drainage images interpreted to be hydro-fracture scarps. **b** Lake 2022-104 (68.56 °N 49.37 °W): drainage by in-lake moulin (yellow circle) identified by the presence of a supraglacial stream within the maximum lake margin that truncates into a moulin located at the maximum lake margin. **c** Lake 2022-178 (68.50 °N 48.90 °W): drainage by overspill identified by the presence of an out-flowing supraglacial stream (yellow arrow) that extends beyond the maximum lake margin. The outflow stream exits the lake's southwestern margin. **d** Lake 2022-157 (68.52 °N 49.02 °W): a frozen lake with no supraglacial exits identified by the lack of outflowing supraglacial streams and a frozen, intact lake-ice surface at the end of the melt season.

## Probabilistic temporal-cluster analysis

Our feature-based catalogue permits clustering investigations for the 11–14% of lakes that drain via hydro-fracture (Fig. 3a, d). We quantify the effect of temporal sampling interval and average hydro-fracture event rate on the probability of observing an event cluster of a given size (Fig. 3b, e). Our approach models event timing as a homogeneous Poisson process[41], asking whether the number of events $m$ observed within the time-observation window $w$ falls above a predicted distribution of event sizes, given an average event rate $\lambda$ (Methods: Probabilistic temporal-cluster analysis).

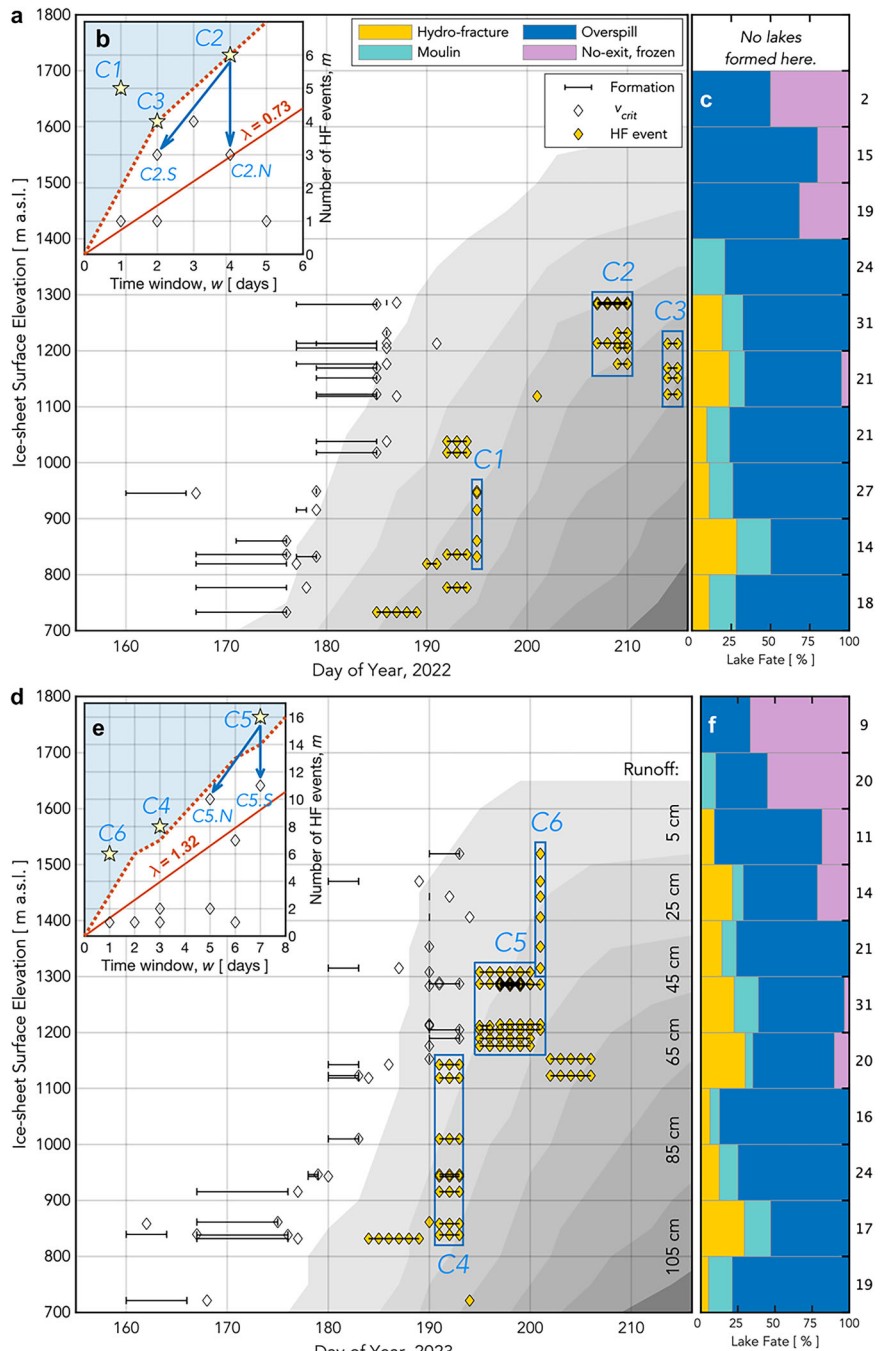

**Fig. 3 | Hydro-fracture event timing constrained by satellite imagery. a** (gold diamonds) All possible 1-d periods of drainage for 22 hydro-fracture (HF) events within the ROI observed over a 30-d window in 2022. Event timing is not further constrained by GNSS observations. Rectangles bound statistically significant clusters. White diamonds show the estimated day by which lakes attain $v_{crit}$, the critical volume required for hydro-fracture; black bars show the time between satellite images bounding initial lake formation. (shading) Daily runoff[38] accumulated at ice-sheet surface elevations within the ROI. **b** (diamonds) Temporal hydro-fracture clusters observed in 2022, with statistically significant clusters

$(P(N_{events} \geq m | \lambda w) < 0.05)$ shown as labelled stars. The red line shows $m$ events expected for time window $w$ given $\lambda$, the observed average daily rate of events. Blue shading with dashed-red-line border denotes *the m–w region of statistically significant clusters*. Blue arrows indicate spatially distinct subclusters within a single, significant temporal cluster. **c** Fate of lakes that formed in 2022, binned by elevation. The number of lakes in each elevation band shown along the right *y*-axis. **d–f** Equivalent to panels (**a–c**) but for 29 HF events observed over a 22-d window in 2023.

In each melt season, we observe three temporal clusters of hydro-fracture events of a statistically significant size given the time-observation window $w$ ($P(N_{events} \geq m | \lambda w) < 0.05$), with the remaining 56–75% of clusters falling within the expected distribution (Fig. 3b, e). These unusually large clusters are possible candidates for triggering interactions promoting drainage. For overspill-type events, comprising two-thirds of the lake dataset, we find only 9–11% of clusters are of a

statistically significant size (Fig. 4e–h). Reading an inter-lake, stress-triggering mechanism into these temporal clusters of overspill events is not physically defensible, given that the process of lakes overspilling their basins is a function of lake-basin geometry and meltwater-inflow rates[23,24,37] and requires no change in ice-sheet stress to occur[12]. For moulin-type drainages, we find 95–100% of temporal clusters fall within the expected distribution (Fig. 4a–d), indicating that

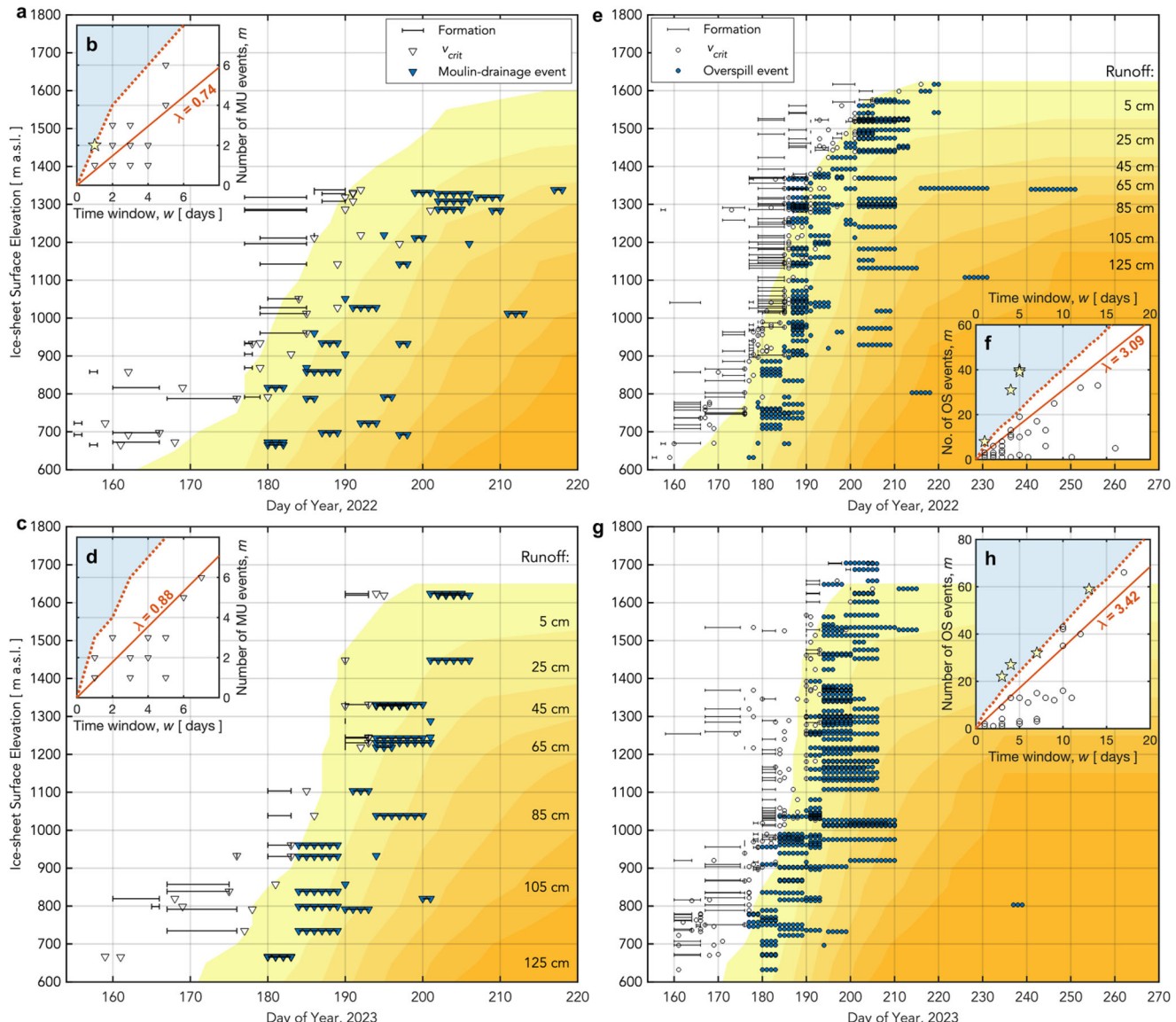

**Fig. 4 | Lake moulin-drainage and overspill event timing constrained by satellite imagery.** Equivalent to Fig. 3, but for the lake-drainage mechanisms of (**a–d**) in-lake moulin drainage and (**e–h**) overspill. **a** All possible 1-d periods of 28 moulin-drainage events (MU) observed over a 38-d window in 2022. White triangles show the estimated day by which lakes attain $v_{crit}$, the critical volume required for hydro-fracture; black bars show time between satellite images bounding initial lake formation. **b** Temporal moulin-drainage clusters observed in 2022, with statistically significant clusters ($P(N_{events} \geq m|\lambda w) < 0.05$) shown as yellow stars. Red line shows

$m$ events expected given the observed average daily rate of events $\lambda$ from 2022/180–218. **c, d** Equivalent to panels (**a, b**) but for 22 moulin-drainage events observed over a 26-d window from 2023/180–206. **e, f** Equivalent to panels (**a, b**) but for 133 overspill (OS) events observed over a 43-d window from 2022/177–220. White circles show the estimated day by which lakes attain $v_{crit}$, the critical volume required for hydro-fracture; black bars show time between satellite images bounding initial lake formation. **g, h** Equivalent to panels (**c, d**) but for 130 overspill events observed over a 38-d window from 2023/177–215.

moulin-type lake-drainage timing is well explained by the average event rate $\lambda$ controlled by elevation-dependent lake-filling rates[38]. Our analysis challenges practices that report temporal clustering without quantifying how temporal aliasing impacts reported cluster size, and without distinguishing hydro-fracture events from overspill and moulin-drainage events[16,17,20,26,31].

When we analyse lake-drainage timing with the knowledge of drainage mechanism, and accounting appropriately for data gaps, we find no evidence for the >100 hydro-fracture events previously reported to occur only over a few days within a similar, southwestern region of the ice sheet[20]. Our study region is located ~150 km north of the southwestern region, and hosts surface lakes at a similar spatial density of 0.027–0.029 lakes km$^{-2}$, compared with 0.028–0.034 lakes km$^{-2}$ (ref. 22). The two ice-sheet regions are analogously devoid of

fast-flowing outlet glaciers whose flow dynamics might impact lake-drainage timing[42]. Because fields of supraglacial lakes form only within a finite range of surface-mass-balance conditions[21,25,43], and within finite ranges of ice-sheet thickness, ice-sheet surface slope, and bed undulation[39,44], our findings are likely broadly applicable to these ice-sheet regions that lie outside the zone influenced by tidewater-terminus dynamics[45].

Our analysis demonstrates that lumping together all lake-volume-reduction events irrespective of lake-volume-loss mechanism (e.g., ref. 20) yields temporal clustering that dominantly tracks overspill-event timing (Fig. 4e–h), and leads to an incorrect inference of long-range hydro-fracture-event triggering as the mechanism responsible for over three quarters of lake-drainage events[20]. The seasonal timing of hydro-fracture event clusters we observe tracks the up-elevation

accumulation of surface runoff (Fig. 3a, d), roughly aligning with existing observations of temporal clusters of neighbouring "rapid" events[8,22,25,34]. However, we underscore that apparent-rapidity methodologies categorising drainage as "rapid" or "slow" provide a poor proxy for lake-drainage mechanism. For example, when following a standard, apparent-rapidity approach that defines near-total lake-volume loss within 96 hours as a "rapid" drainage[8,22,35,46], 11 of the 52 lakes (21%) we observe to have kilometre-scale hydro-fracture scarps in 2022 and 2023 would be sorted into the "slow"-drainage bin because the images bounding these hydro-fracture events are too far apart in time (Fig. 3a, d). Following a stricter, within-48-h apparent-rapidity definition[25] would sort 21 of the 52 hydro-fracture events observed (40%) into the "slow"-drainage bin. Feature-based classification of lake-drainage mechanism, combined with an appropriate accounting for temporal data gaps, avoids these misclassifications, and provides a methodology to move beyond shortcomings of the "rapid"-versus-"slow" analysis framework for supraglacial lake drainage.

## Physical plausibility of event clustering

Some inter-lake, hydro-fracture-event clustering clearly does occur, even if not widespread. Here, we examine the six statistically significant temporal clusters of hydro-fracture events we observe (C1–6; Figs. 5a, 6a) for plausible examples of inter-lake, hydro-fracture-event triggering. We evaluate the plausibility of triggering based on three previously suggested mechanisms: (1) the extent of high tensile stress produced by subglacial blister opening and basal slip[11,12], on elastic timescales of ice deformation (< 1 day); (2) the up-flow extent of tensile stress due to slippery bed patches, on viscous timescales of not-fully-relaxed ice deformation (~ 1 week; ref. 15); and, (3) the propagation rate and pathways of subglacial floods[17,47] (Methods: Physical plausibility of event clustering). Our interrogation of event clustering includes moulin-type drainages, as these events can occur when the lake margin reaches an extant moulin[37] or when high tensile stress produced by hydro-fracture events or passing subglacial floods promotes crevasse opening[17].

We validate our model results empirically by considering whether hydro-fracture occurs in lakes of sufficient volume that lie along potential triggering pathways across the elevation range of interest, either along the ice-flow direction or along subglacial flood routes. In our study region, subglacial-drainage pathways often route water at moderate angles to the ice-flow direction (Fig. 7a; Methods: Subglacial-hydrology modelling), owing to basal ridges that modify the gradient of subglacial hydraulic potential[48,49]. This inland ice-sheet setting differs from that near the margin, where ice-flow direction and subglacial flood pathways often align[16,17,50]. The difference between subglacial flood pathways and the ice-flow direction provides the opportunity to investigate independently the occurrence of hydro-fracture events at lakes connected along ice flow or along the hydraulic gradient. Finally, we ground truth our physics-based evaluation using observations of ice-sheet surface deformation from our GNSS array, which spans 950–1400 m above sea level (a.s.l.) and captures eleven (28%) of the 39 hydro-fracture events of the six clusters (Figs. 5, 6). (Full details of these eleven hydro-fracture events and the six temporal clusters of hydro-fracture events are provided in the Supplementary Information.)

We find plausible examples of inter-lake, hydro-fracture event triggering in all six statistically significant temporal clusters. For example, cluster C1 in 2022 includes five hydro-fracture events all occurring within the same 24 h period (Fig. 5a) and all located along a prominent subglacial discharge pathway (Fig. 7a). The L1A and L1B lakes within the C1 cluster are spatially close enough, and of large-enough volumes, to experience elastic stress coupling via basal blister opening as they drain (Fig. 7b). This interpretation is supported by GNSS observations (Fig. 5b; Supplementary Information Text S1) and by similarity to L1A and L1B hydro-fracture events observed in 2011 and

2013[9,12]. The other three hydro-fracture events in the C1 cluster—L72, L47, and L34—are too far apart to be viable candidates for elastic stress coupling, but are candidates for triggering by the stress change imparted if the blisters propagate (Fig. 7b). Only 24 hr is needed for a subglacial flood propagating at ~ 0.4 m s$^{-1}$ to travel from lakes L1A and L1B to L34, the hydro-fracture event farthest down hydraulic gradient in the C1 cluster (Fig. 7b); thus, we cannot rule out the possibility that a flood event beginning at L1A and L1B propagated beneath, and triggered hydro-fracture initiation at, these three lakes lying down the hydraulic gradient.

By contrast, for some statistically significant temporal clusters, one or two hydro-fracture events are located outside of relevant subglacial pathways and high-tensile-stress regions produced by the other in-cluster events. We consider these hydro-fracture events not to be plausibly related (e.g., lake L2D of cluster C3 in 2022; Fig. 8). Similarly, we split clusters C2 in 2022 and C5 in 2023 into two spatially independent subclusters because the subclusters occur 15–20 km apart in the across-ice-flow direction, with the region separating the subclusters hosting 12 (2022; Fig. 5a) to 16 (2023; Fig. 6a) lakes that do not drain via hydro-fracture during the time of the cluster drainages. Similarly, the modelled extents of high-tensile-stress regions from the C2 and C5 events show a ~ 15-km-stretch of no elevated surface stress between the subclusters (Supplementary Information Figs. S2.1b, S5.1b). Moreover, the C2 and C5 subclusters are unlikely to be related through subglacial pathways, with southern subclusters' subglacial floods propagating westward along subglacial lows to beneath MHIH (Fig. 1b, c and Supplementary Information Figs. S2.2c, S5.2c), and northern subclusters' subglacial floods likely propagating either northwestward, to regions north of the GNSS array, or southwestward, to beneath the GNSS-instrumented basins at 1150 m a.s.l. (Fig. 1b, c and Supplementary Information Figs. S2.1a, S5.1a). The sizes of these four subclusters lie within expected cluster-size distributions (Fig. 3b, e). That statistically significant temporal clusters contain hydro-fracture events at similar altitudes—yet spaced too far apart to be plausibly related via inter-lake, hydro-fracture-event triggering—exemplifies the temporal clustering of lake drainage that arises stochastically due to elevation-dependent lake-filling rates[12,25,34].

Comparing across the two melt seasons, we observe interannual variability in the lake-drainage mechanism for some lakes, with the group of instrumented lakes at 1150 m a.s.l. (L2A–C) exhibiting variability in the capacity for inter-lake, hydro-fracture event triggering to occur in the inland direction (Figs. 5a, 6a). In 2022, lake L2C drains via hydro-fracture within a ~ 2-hr, GNSS-constrained time window of the L2A and L2B hydro-fracture events (Figs. 5, 8a and Supplementary Information Text S3), yielding a potential example of inter-lake, hydro-fracture event triggering via the slippery-patch mechanism[15]. Lake L2C is located ~5 km inland of L2A, and thus would likely fall within the tensile-stress region caused by an L2A slippery-patch that has an along-ice-flow length equal to twice the L2A idealised-blister diameter (Fig. 8b). In 2023, lakes L2A and L2B hydro-fracture at lower volumes and lake L2C freezes over at the end of the melt season (Fig. 6a), perhaps because the stress perturbation at L2C will be smaller when the subglacial blister and slippery-patch sizes at L2A and L2B are smaller.

We find a small number of moulin-drainage events to be associated with half of the C1–6 clusters in space and time, bringing the size of plausibly triggered events for all six clusters to 2–7% of the total lakes viable for hydro-fracture at the time of cluster occurrence (Supplementary Information Table S8). In both years, dozens of lakes of sufficient volume to hydro-fracture to the bed are located immediately up- and down-ice-flow from the hydro-fracture events, but drain by overspill or refreeze without draining (e.g., Figs. 7, 8). The absence of hydro-fracture in these basins observationally supports a rapid falloff in stress-perturbation magnitude with inland distance[15], and is consistent with downstream subglacial water routing through

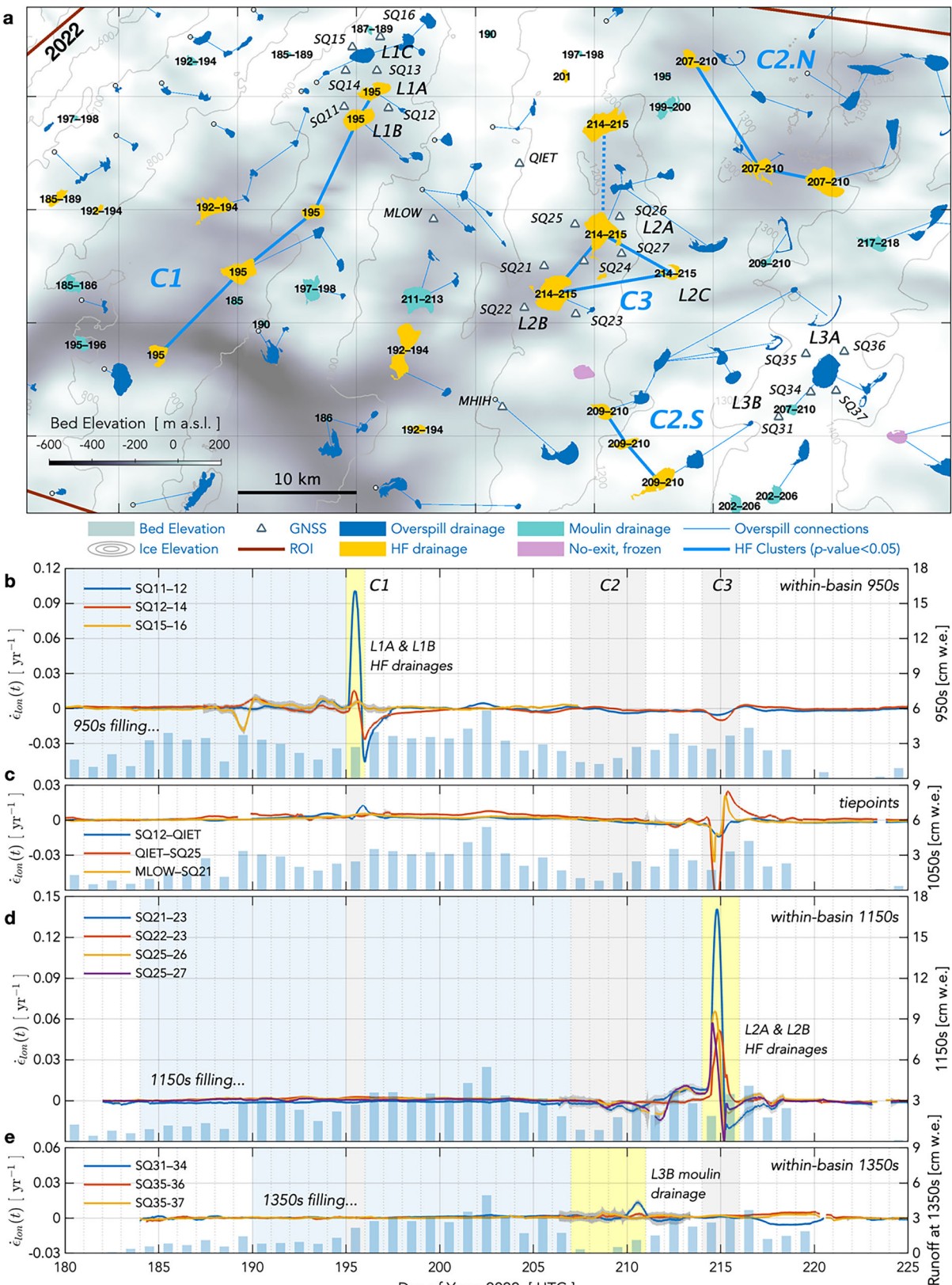

**Fig. 5 | Hydro-fracture-event clusters and strain rates observed in 2022.**
**a** Supraglacial lakes and their (legend) drainage mechanisms and overspill con-
nections. Days of the year of hydro-fracture and moulin-drainage events shown
with black numbers. Statistically significant ($P(N_{events} \leq m|\lambda w) < 0.05$) temporal
clusters of HF events linked with heavy, blue-dashed lines and labelled C1–3. Phy-
sically plausible, inter-lake triggered events linked with heavy, solid-blue lines.
**b**–**e** Along-flow horizontal strain rates $\dot{\varepsilon}_{lon} \pm 3\sigma$ uncertainty bands (dark grey

envelopes) between GNSS-station pairs shown as anomalies relative to the value on
2023/165.0 prior to the onset of daily runoff[38] (blue bars, right $y$-axis). Blue shading
shows the duration of lake presence. Yellow shading shows day(s) of hydro-fracture
clusters that include drainages of lakes within GNSS-station clusters at elevations
near 950, 1150, and 1350 m a.s.l. Light grey shading shows the timing of hydro-
fracture clusters in other regions.

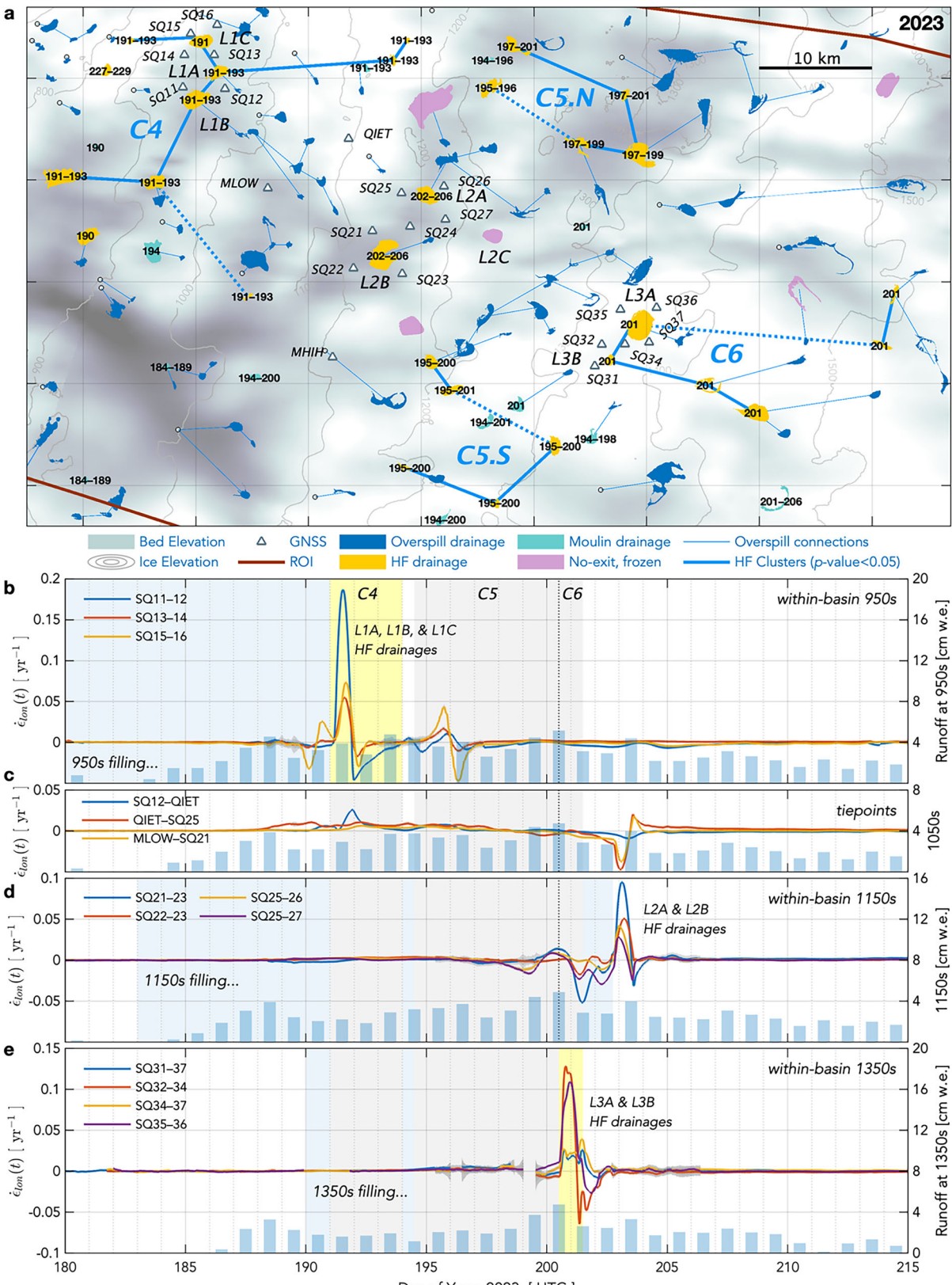

**Fig. 6 | Hydro-fracture-event clusters and strain-rate transients observed in 2023.** Equivalent to Fig. 5, but for the 2023 melt season. **a** Supraglacial lakes and their (legend) drainage mechanisms and overspill connections. Statistically significant ($P(N_{events} \leq m | \lambda w) < 0.05$) temporal clusters of HF events linked with heavy, blue-dashed lines and labelled C4–6. Physically plausible, inter-lake triggered events linked with heavy, solid-blue lines. **b**–**e** Along-flow horizontal strain rates $\dot{\varepsilon}_{lon} \pm 3\sigma$ uncertainty bands (dark grey envelopes) between GNSS-station pairs shown as anomalies relative to their value on 2023/165.0. Dashed vertical lines in panels (**b**–**d**) show separation between hydro-fracture-event clusters C5 and C6 in time.

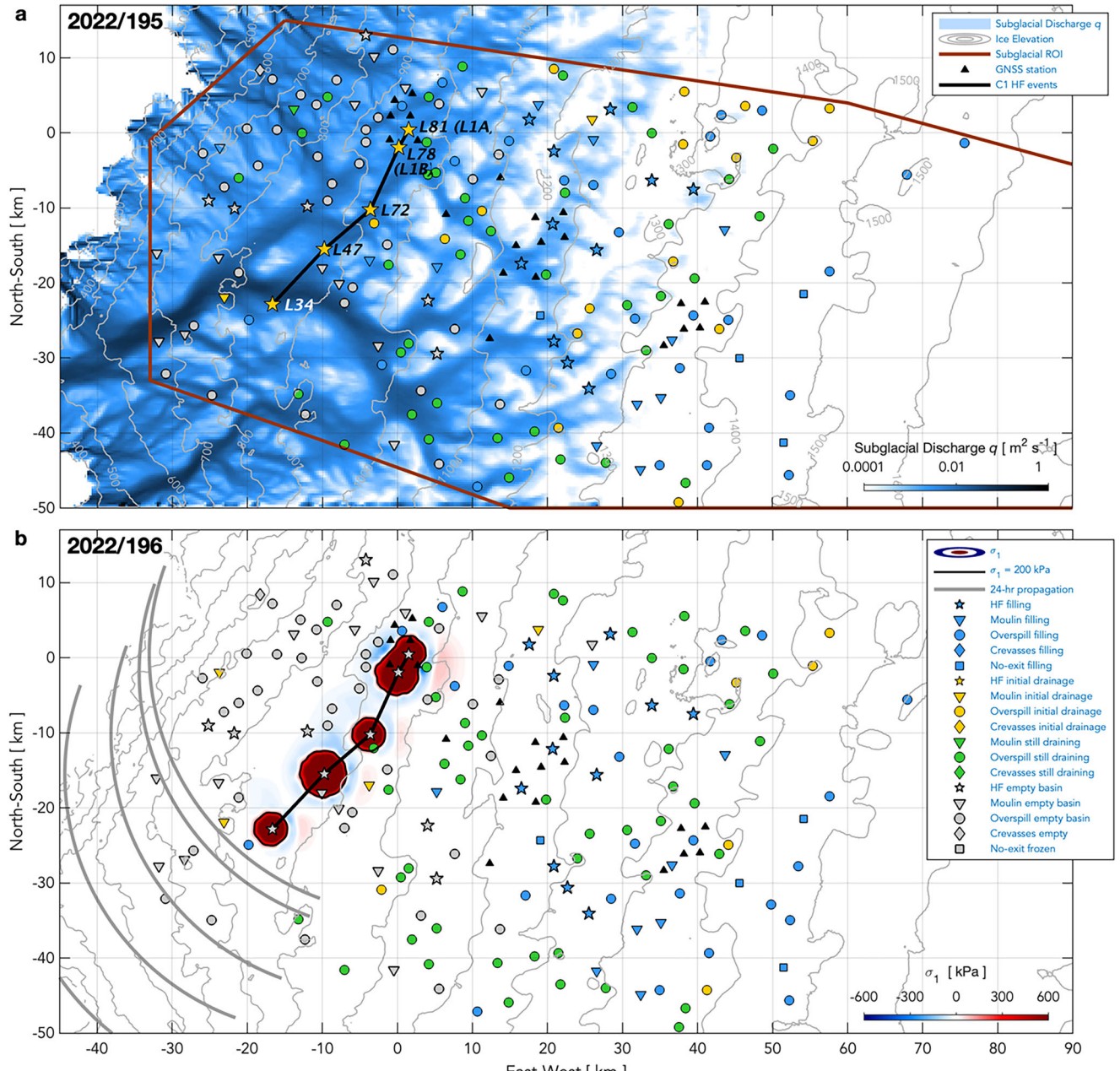

**Fig. 7 | Modelled subglacial discharge and ice-sheet surface stress during hydro-fracture event cluster C1 (2022/195). a** Modelled subglacial discharge on 2022/195. Brown line outlines region of interest (ROI). Five C1 hydro-fracture events labelled with their identification numbers and linked with black lines. Grey contours show ice-sheet surface elevation. **b** Modelled maximum principal stress $\sigma_1$ from blister opening and basal slip. Black lines show 200-kPa contour in $\sigma_1$; red shading shows an increase in tensile stress, and blue shading shows increased compression. Cuspate lines show blister-propagation front 1 d past time of drainage, using flood-propagation aspects of 200–290° from the C1 events. Symbol shapes show the lake-drainage mechanism and symbol colours show lake status on (**a**) 2022/195 and (**b**) 2022/196. Map origin is the 2011 position of the "North Lake" M1 moulin (68.72 °N 49.53 °W) identified in past work[9,12].

constrained flood-propagation routes, rather than as extensive, sheet-like flow[12,49–51].

When we analyse lake-drainage timing with the knowledge of drainage mechanism, lake location, and subglacial-routing pathways, we find no evidence for long-range, inland triggering of hydro-fracture events, and no evidence that the majority of lakes drain via triggered hydro-fracture. The only plausible triggering events we identify that connect drainages outside a single elevation band are a small number of possible interactions between hydro-fracture and moulin-drainage events, whose collective numbers amount to a small percentage of all viable lakes. These low numbers may still overestimate the true proportion of lakes that drain via inter-lake triggering mechanisms: we

cannot rule out that the coincident drainage timing of some tallied events arises purely from elevation-dependent lake-filling rates (e.g., the five C1 hydro-fracture events; Fig. 5a).

## Hydro-fracture across the ablation zone
The hypothesis that lake drainage via hydro-fracture accelerates grounded ice-flow destabilisation during past[2], present[3], and future[4–7] climate-warming scenarios hinges on applications[14] of theoretical work[13,15] arguing that ice-sheet acceleration lower in the ablation zone can drive hydro-fracture events and the creation of new, surface-to-bed meltwater pathways beneath inland supraglacial lakes[20]. This theory predicts above-background, extensional strain rates across

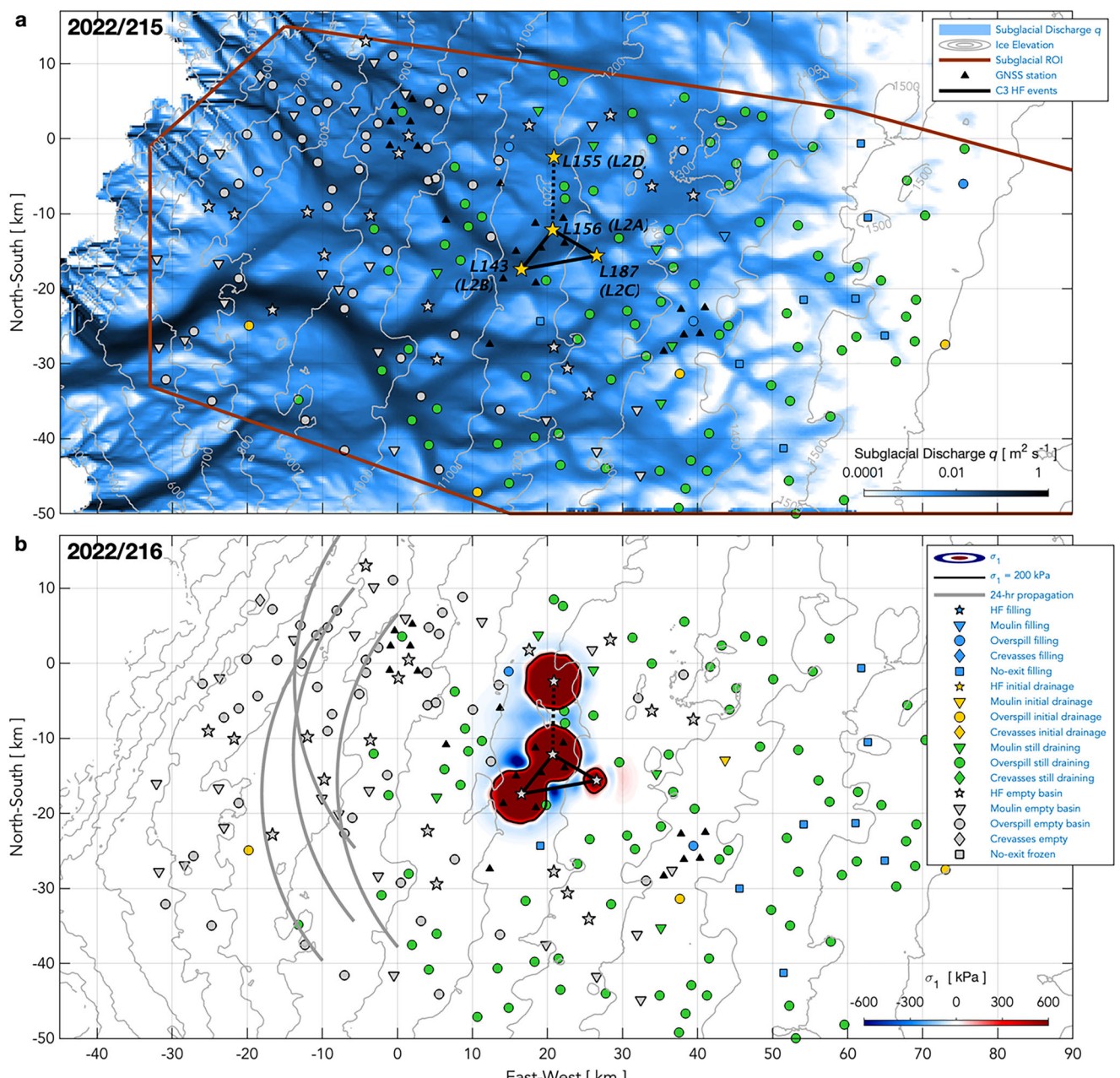

**Fig. 8 | Modelled subglacial discharge and ice-sheet surface stress during hydro-fracture event cluster C3 (2022/214–215). a** Modelled subglacial discharge on 2022/215. Brown line outlines region of interest (ROI). Four C3 hydro-fracture events labelled with their identification numbers and linked with black lines. Statistically significant ($P(N_{events} \leq m | \lambda w) < 0.05$) temporal clusters of hydro-fracture events linked with black-dashed lines; physically plausible, inter-lake triggered events linked with solid-black lines. Grey contours show ice-sheet surface elevation. **b** Modelled maximum principal stress $\sigma_1$ from blister opening and basal slip. Black lines show 200 kPa contour in $\sigma_1$; red shading shows an increase in tensile stress, and blue shading shows increased compression. Cuspate lines show blister-propagation front 1 d past time of drainage, using flood-propagation aspects of 230–310° from the C3 events. Symbol shapes show lake-drainage mechanism and symbol colours show lake status on (**a**) 2022/215 and (**b**) 2022/216. Map origin is the 2011 position of the "North Lake" M1 moulin (68.72 °N 49.53 °W) identified in past work[9,12].

inland lake basins as the melt season progresses at lower elevations[14], and, most especially, during times of lower-elevation hydro-fracture events[20].

During both melt seasons we observe, ice-sheet velocities gradually increase at all seven GNSS-instrumented lake basins as the melt season begins. During this period, ice-sheet velocities bracketing the sets of lakes increase largely in tandem, resulting in negligible longitudinal strain rates across the basins (Figs. 5b–e, 6b–e), with perturbations in longitudinal strain rates between GNSS station pairs > |0.002| yr$^{-1}$ detectable given GNSS measurement precision

(Methods: GNSS-derived quantities). Longitudinal strain-rate excursions $\pm$ 0.02 yr$^{-1}$ in magnitude commence 2–6 d prior to hydro-fracture within each lake set, likely indicating local basal-water movement as surface-to-bed meltwater pathways open in nearby regions. For example, a moulin-lake drainage event occurs immediately north of lake L1C approximately six days prior to the 2022 L1A and L1B hydro-fracture events (Fig. 5a); this moulin-lake drainage event drives strain-rate excursions SQ15–SQ16 and between SQ12–SQ14 that do not trigger the drainage of L1A, L1B, or L1C (Fig. 5b). Similarly, in 2022 and 2023, lake hydro-fracture and moulin-drainage events inland of

L2A and L2B drive strain-rate excursions across both of these basins that commence about five days prior to the eventual hydro-fracture of L2A and L2B (Figs. 5d, 6d). These pre-hydro-fracture strain excursions, which do not themselves result in hydro-fracture, indicate that lake basins are resilient to some low-level strain-rate transients in the lead-up to hydro-fracture initiation. Finally, longitudinal strain-rate excursions of $+0.02\,\mathrm{yr^{-1}}$ to $+0.18\,\mathrm{yr^{-1}}$ occur during the hydro-fracture events themselves. As the melt season continues, lower-magnitude strain-rate excursions coincide with high-surface-runoff periods and passing subglacial floods.

Our regional, mechanistic lake-drainage catalogue allows us to interpret the GNSS-observed, strain-rate response of inland basins to lower-elevation hydro-fracture events. In both years, multiple hydro-fracture events instigate subglacial flooding across a width of ~30 km (C1; Fig. 5a) and ~18 km (C4; Fig. 6a) of the 850–950 m a.s.l. elevation band. Meanwhile, rates of extension across our GNSS-instrumented lakes at 1150 and 1350 m a.s.l. are unperturbed, and tens of viable lakes sitting up-ice-flow at 900–1100 m a.s.l. fail to drain via hydro-fracture (Figs. 5d, e, 6d, e). Two "tiepoint" GNSS stations, QIET and MLOW, located within the 1000–1100 m a.s.l. elevation band, allow us to investigate hydro-fracture-induced strain-rate excursions for the ice-sheet region between the 950 and 1150 m a.s.l. GNSS-instrumented basins (Figs. 5a, 6a). Here, hydro-fracture-induced speed-ups at SQ12 result in positive strain-rate excursions between SQ12 and QIET (Figs. 5c, 6c), though the magnitudes of these strain-rate excursions are small ($<0.025\,\mathrm{yr^{-1}}$), consistent with a $+0.015\,\mathrm{yr^{-1}}$ longitudinal strain-rate perturbation observed up to ~4 km inland of three lower-elevation, lake-drainage events in the Paakitsoq region[16]. Moreover, none of the five or so overspill-type lakes that form in between SQ12 and QIET drain via hydro-fracture in either year. Rates of extension between the two tiepoint stations and the GNSS stations at 1150 m a.s.l. are unperturbed (i.e., below the measurement-detection threshold of $\pm 0.002\,\mathrm{yr^{-1}}$) as subglacial-flood events transpire in the 850–950 m a.s.l. elevation band (Figs. 5c, 6c). The absence of a strain-rate response across inland lake basins while broad-scale subglacial-flood events transpire across lower-elevation regions is evidenced a second time in each year by unperturbed longitudinal strain rates across GNSS-instrumented basins at 1350 m a.s.l. while 2–4 hydro-fracture events occur within GNSS-instrumented basins at 1150 m a.s.l. (Figs. 5e, 6e).

The absence of a strain-rate response within higher-elevation lake basins during hydro-fracture events in lower-elevation regions argues in support of a small limit, of a few ice thicknesses, on the distance over which lower-elevation ice-flow modulation can promote hydro-fracture initiation at higher elevations. The existence of dozens of overspill-type lakes of sufficient volume to hydro-fracture, located immediately inland and down-flow of the hydro-fracture event clusters (Figs. 5a, 6a), empirically confirms the short distance over which inter-lake, hydro-fracture event triggering takes place. Complex spatial patterns of ice-sheet surface stress observed immediately following hydro-fracture events constrained by denser GNSS arrays[12]; basal ridges separating lake-forming regions[49]; and hypothesised narrow subglacial flood widths ($<2\,\mathrm{km}$) that follow bed topographic lows[12,51,52] may all help explain why many candidate lakes viable for hydro-fracture do not, in fact, fracture, and instead drain via overspill, or freeze over.

The elevation of the highest 5% of lakes in our study area has climbed 67 m a.s.l. decade$^{-1}$ during the last forty years[21], making lake drainage at higher elevations possible should ice-sheet-stress and surface-melt conditions be favourable. We find that the proportion of hydro-fracture and moulin-drainage events decreases with elevation, leaving the highest-elevation regions comprised of overspilling and no-exit, frozen lakes (Fig. 3c, f). We observe interannual variability in hydro-fracture drainage in upper-elevation regions in our study's two-season snapshot, with the uppermost hydro-fracture events located 200 m higher in 2023 than in 2022. The uppermost 2023 hydro-

fracture events occur during a runoff season that exhibits faster up-elevation runoff onset (Fig. 3a, d), suggesting that whether locally sourced surface melt can access the ice-sheet bed before subglacial-drainage systems attain much efficiency may be an important control on upper-elevation hydro-fracture initiation, possibly due to the greater impact on subglacial water pressures that arises from melt-water injection into the low-efficiency subglacial drainage systems[53–55] expected beneath thick ice[56]. This interpretation aligns with the 2–6-d periods of strain-rate fluctuation we observe immediately preceding hydro-fracture initiation across all elevations (Figs. 5, 6). Moreover, the near-annual occurrence of altitudinally extensive, late-season melt events[57,58] does not produce hydro-fracture of the remaining, undrained high-elevation lakes, further supporting the hypothesis that early-melt-season runoff characteristics are a key environmental control on high-elevation hydro-fracture initiation.

Across the ablation zone, we find hydro-fracture initiation to be driven by a combination of local, largely early-season, meltwater access to the bed and interactions between nearby hydro-fracture events. Our observation that strain rates across inland lake basins are unperturbed during extensive subglacial-flood events in lower-elevation regions argues for a simple model of inland hydro-fracture advance during periods of climate warming: inland evolution of surface-to-bed meltwater pathway formation beneath lakes migrates alongside advancing surface melt, but is not accelerated by drainage activities at distant, lower-elevation lakes.

## Methods
### Mechanistic lake-drainage catalogues
We track the drainage mechanism and timing of supraglacial lake-drainage events in melt seasons 2022 and 2023 using daily, composite 10-m-resolution Sentinel-2 optical images and 25-m-resolution Sentinel-1 Synthetic Aperture Radar (SAR) images of an ~11,600 km$^2$ ice-sheet region. This region extends ~700–1700 m above sea level (a.s.l.) (Fig. 1). Composite images are downloaded for all days when lakes are visible—even if there exists a high percentage of cloud cover—resulting in 46 Sentinel-2 images from 2022/153–276 (inclusive) and 41 Sentinel-2 images from 2023/159–276 (inclusive); four additional Sentinel-1 SAR images are available for each melt season. Over our ice-sheet region, Sentinel-2 images are consistently taken between ~15:00–15:30 UTC and all eight Sentinel-1 images used are taken between 20:44–20:49 UTC. Using the Fully Automated Supraglacial lake Tracking at Enhanced Resolution (FASTER) algorithm[30], we identify lake locations and track lake surface area for lakes that surpass a minimum lake-surface-area threshold of 0.165 km$^2$ (ref. 30). We identify a total of 342 potential lake locations in 2022 and 406 potential lake locations in 2023, which we independently inspect for timing of lake formation, timing of lake draining, and lake-drainage features, using all available images of each lake basin.

After removing FASTER-identified potential lake locations that are streams, slush features, water-filled crevasses, and supraglacial lakes truncated by the imagery boundary, we visually log image dates of initial lake formation, maximum lake surface area, last pre-drainage basin, initial post-drainage basin, and initial frozen lake surface. We further classify each lake by its drainage mechanism through visual identification of lake-drainage features (Fig. 2). Lake drainage by hydro-fracture is identified through the presence of bright, linear, kilometre-scale features in immediate, post-drainage images that we interpret as hydro-fracture scarps; the truncation of inflowing supraglacial streams along these identified hydro-fracture scarps; the presence of bright, tens-of-metres-scale features in post-drainage basins that we interpret as ice blocks[1] fractured off of the main hydro-fracture plane; and, finally, the lack of outflowing supraglacial streams from the post-drainage basin. Lake drainage by moulin is identified by the presence of a moulin located at, or within, the margin of the maximum lake extent. Lake drainage by overspill, by contrast, is identified by the presence of one or

multiple outflowing supraglacial streams that do not truncate at a moulin located within the margin of the lake's maximum extent. The fate of lateral meltwater movement for each overspilling lake is traced to a downstream supraglacial lake or moulin (Fig. 1b, c). The final category we denote is a frozen lake with no supraglacial exits, which we identify by the lack of outflowing supraglacial streams and a frozen surface of the lake at the end of the melt season.

During our spatiotemporal assessment of hydro-fracture event clusters, we assess only the lakes that form within the 7,189-km$^2$, subglacial-catchment-delineated region of interest (ROI) encompassing our GNSS array (Fig. 1b, c). To assess only the lakes that are viable for hydro-fracture, we estimate the day of the year by which each lake attains a critical volume required for hydro-fracture. This critical volume threshold $v_{crit}$ is defined as the volume required to keep a vertical crack entirely water-filled until the crack reaches the ice-sheet bed[59]. We model a timeseries of volume for each lake by linearly interpolating lake volume from an assumption of zero volume on the day immediately prior to the image date of initial lake formation, up to the image date of the lake's maximum, FASTER-derived estimate of volume (Figs. 3, 4). We approximate this maximum lake volume by calculating an average lake diameter from the FASTER-derived lake surface area estimate, and then assume a conical lake with a diameter-to-lake-depth ratio of 100:1 (ref. 40). We calculate the hydro-fracture depth potential of the timeseries of interpolated lake volumes using the ice-sheet thickness beneath the lake and assuming a glaciological setting of neutral differential ice stress and a mid-range shear modulus for glacial ice of 1.5 GPa (ref. 40). A lake is a candidate lake for hydro-fracture on all days after its volume exceeds $v_{crit}$ (for its local ice-sheet thickness) and before the lake has drained.

### Probabilistic temporal-cluster analysis
We model lake-drainage event rates as a homogeneous Poisson process, asking whether the number of events observed over a given time interval falls outside of the distribution of events predicted, given the average rate of events observed[41]. For a homogeneous Poisson process, the probability of observing exactly $m$ events in a time period $w$, given an average event rate $\lambda$, is:

$$P\left(N_{events} = m | \lambda w\right) = \frac{(\lambda w)^m e^{-\lambda w}}{m!}. \tag{1}$$

Since the temporal resolution of our satellite-observed drainage dates is, at best, 1 d, we define the average event rate $\lambda$ as the number of events within the ROI, divided by the total number of possible 1-d periods during which the events occurred. The average event rate of drainages via hydro-fracture is 0.73 drainages d$^{-1}$ in 2022 (i.e., 22 hydro-fracture events observed in a 30-d window; Fig. 3b) and 1.32 drainages d$^{-1}$ in 2023 (i.e., 29 hydro-fracture events observed in a 22-d window; Fig. 3e). In 2023, there is one late-melt-season hydro-fracture event (2023/227–229; Fig. 1c) that begins 21 d following the cessation of the time window of the other 29 hydro-fracture events that year (2023/184–206). We exclude this one late-melt-season hydro-fracture event from the calculation of $\lambda$ in 2023, given the marked difference between event rates in these two halves of the 2023 melt season.

We identify all unique temporal clusters of hydro-fracture events, with cluster size defined as the number of events within the temporal window between sequential images available for constraining each lake's hydro-fracture event. For example, if the hydro-fracture event date for one lake is constrained to 4 d, the cluster size for that hydro-fracture event will be that event plus all other hydro-fracture events that occur completely within those same 4 d. Finally, due to the small number of hydro-fracture events observed, we approach this cluster analysis from an a priori assumption of no elevation control on hydro-fracture timing. However, in reality, lake-drainage timing via hydro-fracture in our study region closely follows elevation-dependent

surface melt rates that also govern when lakes form and attain critical lake volumes required for hydro-fracture[37] (Fig. 3a, d). We conduct the equivalent temporal-clustering analysis for lake moulin-drainage and overspill events within the ROI (Fig. 4).

### GNSS data
We deployed a network of 23 geodetic-quality, multi-frequency Xeos Resolute GNSS receivers operating with Septentrio PolaNt-x antennas to log continuously at a 5-s interval from late May 2022 through mid-September 2023. Antennas were mounted on aluminium poles initially drilled ~3.5 m into the ice such that the antenna base sat immediately above the snow surface at installation, with antenna distance above the surface increasing during the ablation season. Array geometry targeted pairs or triplets of lakes draining sequentially via hydro-fracture[12] at ~950 m a.s.l. (lakes L1A–C; GNSS stations SQ11–16), ~1150 m a.s.l. (lakes L2A, L2B; GNSS stations SQ21–27), and ~1350 m a.s.l. (lakes L3A, L3B; GNSS stations SQ31–37); two moulins along subglacial-flow pathways (MLOW, MHIH); and a region relatively free from surface-to-bed meltwater pathways, lakes, and their drainages (QIET) (Fig. 5a). The network spans an along-flow distance of ~55 km from the mid- to the upper ablation zone (950–1400 m a.s.l.). The onset of ice slabs at this latitude (68–69° N) occurs at ~1600 m a.s.l.[43], and the long-term elevation of the equilibrium line altitude (ELA), calculated using the 1-km-resolution RACMO surface mass balance product averaged from 1958–2019, sits at ~1700 m a.s.l.[38] At times during the 17-month deployment, monument and/or receiver malfunction resulted in attrition of stations SQ32 and/or SQ33, such that the network consisted of 21 stations in melt season 2022 and 22 stations in melt season 2023.

Global Positioning System (GPS) data were processed in kinematic mode using the ionosphere-free $L_C$ phase observable in the TRACK (v. 1.53) module[60] of the GAMIT/GLOBK (v 10.71) software package[61] relative to the Greenland GNSS Network base station KAGA[62], located on bedrock at 50–95 km from the on-ice stations, to yield position estimates every 15 s. We used precise satellite orbits and clocks from the International GNSS Service. Data with elevation angles below 5° were excluded. With motion at some sites being rapid (~20 m d$^{-1}$) during the hydro-fracture events, we processed all of the data with a random-walk parameter of 1.155 ms$^{-1/2}$ to produced lightly constrained solutions. We estimated atmospheric parameters to avoid mapping atmospheric variability into vertical positions, but we did not model ionospheric effects not eliminated by the $L_C$ combination.

To reduce offsets at day boundaries, we processed the data in 36-hr segments, with six hours of data extending beyond the day, and selected the central 24 hr of position estimates for each day to produce a daily kinematic solution timeseries. We eliminated estimates with more than two unfixed biases (~3% of estimates); or with position estimates >3$\sigma$ from the mean value calculated over a sliding, 6-hr window centred on the data point (~3% of estimates); or with formal errors (one standard deviation) larger than 0.0425 m in the vertical position (24% of estimates in 2022 and 30% of estimates in 2023). Rejecting a third of estimates was required to remove artificial diurnal periodicity in horizontal and vertical positions sourced primarily from periodic, multipath noise. The ice-sheet surface represents a high-multipath environment, particularly as the melt season progresses and the surface becomes wetter and rough on multiple scales. The data-acceptance criteria do not affect interpretations of strain-rate changes across upper-elevation lake basins when lower-elevation lakes drain. One-standard-deviation errors on the final, selected horizontal and vertical positions are typically 0.02 m and 0.04 m, respectively, across both melt seasons.

### GNSS-derived quantities
We use GNSS surface-position estimates to quantify strain-rate transients (Figs. 5, 6) and bed separation (Supplementary Information)

instigated by runoff inputs to the ice-bed interface and travelling subglacial floods. To estimate the ice-sheet surface horizontal and vertical velocities needed for calculating these quantities, we use centred, sliding least-squares regression of surface-position estimates. Periodic, multipath noise in the horizontal and vertical position estimates results in artificial diurnal periodicity in horizontal and vertical velocities for sliding least-squares regression window widths < 30 hr; however, a wide window width poorly characterises ice-flow acceleration during hydro-fracture events. Thus, two window widths are used in the calculation of velocity, with an 18-hr window used during days of lake-drainage events and a 36-hr window width used outside of these times. In 2022, an 18-hr window is used from 2022/187–199 at the 950s, 2022/206–217 at the 1150s, 2022/206–212 at the 1350s, 2022/212–215 at MLOW, and 2022/206–212 at MHIH (Supplementary Information Fig. S0.1). In 2023, an 18-hr window is used from 2023/188–196 at the 950s, 2023/195–205 at the 1150s and 1350s, and 2023/193–205 at MHIH (Supplementary Information Fig. S0.2). Velocities are presented for 18-hr time windows when at least 6-hr-worth of surface-position estimates are available; velocities are presented for 36-hr time windows when at least 12-hr-worth of surface-position estimates are available.

Following ref. 12, 30-min-resolution horizontal strain rates between GNSS stations are calculated from surface-position estimates and 30-min-resolution velocities in along- and across-flow directions as defined by station clusters' mean-flow directions in pre-runoff time periods. The mean along-flow directions of each station cluster during the Spring, pre-runoff time periods of 2022/150–160 and 2023/150–160 are calculated as the direction of horizontal velocities averaged over the 6–7 stations within each cluster. The mean along-flow directions in Spring 2022 and 2023, respectively, are 277° and 277° for the 950s; 266° and 266° for the 1150s; and 278° and 277° for the 1350s. We present along-flow horizontal strain rates between GNSS-station pairs $\dot{\varepsilon}_{lon}$ as anomalies relative to the along-flow horizontal strain rate between the station pair on 2022/165 or 2023/165 to isolate strain-rate transients driven by runoff inputs and lake-drainage events.

Following ref. 12, $1\sigma$ errors in along-flow horizontal strain rates between GNSS-station pairs $\delta\dot{\varepsilon}_{lon}$ are calculated via arithmetic error propagation of error estimates of station velocities and station positions. Estimates of $\delta\dot{\varepsilon}_{lon}$ for different GNSS-station pairs range from $7 \times 10^{-5}$ to $5 \times 10^{-4}$ yr$^{-1}$, when averaged in time from 2022/165–230 and from 2023/165–230 (Supplementary Information Table S10). We take strain-rate perturbations three times greater than the upper end of our $\delta\dot{\varepsilon}_{lon}$ estimates ($5 \times 10^{-4}$ yr$^{-1}$) to be detectable given measurement precision, placing the bound on a detectable strain-rate perturbation at $\pm 0.002$ yr$^{-1}$ for the dataset. With $\delta\dot{\varepsilon}_{lon}$ estimates dependent on inter-station distance and the quality of the velocity estimate, we present $\pm 3\sigma$ error envelopes for $\dot{\varepsilon}_{lon}$ timeseries on all relevant figures (Figs. 5, 6 and Supplementary Information Figs. S1.2, S2.2, S3.2, S4.2, S5.2, and S6.2). Sharp changes in $\pm 3\sigma$ error envelopes on $\dot{\varepsilon}_{lon}$ estimates occur when switching between 36-hr and 18-hr window widths in the calculation of velocity, with 18-hr window width velocities having larger errors due to the smaller number of points within the sliding least-squares regression of surface-position estimates used to calculate the velocity estimate.

### Subglacial-hydrology model

We estimate the location, discharge, and onset timing of subglacial drainage pathways in our study region using a numerical model of subglacial hydrology with cavity and channel components[53] for realistic ice and bed geometries[48,55] (Figs. 7a, 8a). We force the model by distributing daily RACMO runoff estimates[38] on 300-m-spaced model grid nodes that align with every other grid point of the 150-m-spaced ice-sheet surface and bed elevation model BedMachine v.5[63,64]. Additional melt inputs to the drainage system include a constant rate of basal melt from geothermal-heat fluxes and basal melt from frictional sliding at

the ice-bed interface, which we calculate using a static, spatially non-uniform basal sliding velocity[55] set to surface-velocity values observed by the 2022 MEaSUREs Annual Velocity Mosaic[65]. Parameter-value choices for this model have been previously constrained for this region[48]; values of key parameters include: englacial void fraction ($10^{-3}$), cavity sheet permeability ($10^{-3}$ Pa$^{-1}$ s$^{-1}$), width of the cavity sheet that contributes to channel melting (1000 m), and basal-undulation height (0.1 m) and length (10 m) scales.

We use model-estimated subglacial-discharge pathways to guide our interpretations of flood-propagation pathways (Figs. 7, 8 and Supplementary Movies 1, 2). Though a high priority for the ice-sheet hydrology community is realistically including the impact of hydro-fracture-driven lake drainages on subglacial drainage system evolution, we leave that advance to future work and do not force the subglacial-hydrology model with the volume and timing of hydro-fracture events within the lake-drainage catalogues. Finally, as there is no surface storage of runoff in the model, we interpret subglacial drainage system onset timing as the earliest possible establishment of coherent surface-meltwater routing at the bed.

### Physical plausibility of event clustering

We interrogate whether the temporal hydro-fracture-event clusters observed are plausible examples of inter-lake, hydro-fracture-event triggering using three different modelling considerations: (1) the extent of high-tensile-stress regions produced by subglacial blister opening[11] and basal slip, on elastic timescales of ice deformation (< 1 day)[12,66]; (2) the up-ice-flow extent of tensile-stress regions due to slippery bed patches, on viscous timescales of not-fully-relaxed ice deformation (~1 week)[15]; and (3) the propagation rate of subglacial blisters travelling down flood path[17,47,55].

For modelling consideration (1), following refs. 11,12, we forward model ice-sheet surface stresses produced by idealised basal-cavity opening[11] and basal-slip[12] distributions for hydro-fracture-event lake volumes to estimate the spatial extent of high-tensile-stress regions during the initial day of the event. We define a radial blister[11] of the maximum, FASTER-estimated lake volume. The height of this blister defines the amount of basal-cavity opening at a depth of the ice-sheet thickness within an elastic halfspace[12]. At this same depth within the halfspace, we impose 0.5 m of bed-parallel slip in the down-ice-flow direction within the circumference of the blister[12]. Ice-sheet surface deformation due to this prescribed basal-cavity opening and slip is forward modelled using the Okada (1985) Green functions[66], and the modelled surface-strain components are presented as maximum principal stress $\sigma_1$ (Figs. 7b, 8b), using mid-range material properties for glacial ice of a Poisson ratio of 0.3 and a shear modulus of 1.5 GPa[40]. We use a sign convention that $\sigma > 0$ indicates tension. Further details and illustrations on the contribution to ice-sheet surface stress from the individual components of basal-cavity opening and basal slip are provided in previous work for a range of lake-drainage volumes and ice-sheet thicknesses[12] that encompass the volume and ice-thickness of hydro-fracture events within the 2022 and 2023 lake-drainage catalogues.

While modelling consideration (1) uses the elastic stress change generated by hydro-fracture events to evaluate the plausibility of triggering (e.g., Fig. 7b), we note that these elastic stress changes take place in the context of background, viscous stresses driven by the ice-sheet geometry[9,12]. Maximum principal stresses of these background stresses at the lake locations range from −44 to +148 kPa when estimated using the Glen flow law[67] to convert ice-sheet surface strain rates calculated from the 2022 and 2023 MEaSUREs Annual Velocity Mosaics[65] to ice-sheet surface stresses[9,12]. For this calculation, we use a flow-law creep parameter $A$ value of $3.5 \times 10^{-25}$ s$^{-1}$ Pa$^{-3}$ recommended for −10 °C ice temperatures[68,69].

For modelling consideration (2), we estimate the average surface tensile stress induced by slippery ice-sheet bed patches produced by

the 39 in-cluster hydro-fracture events. We assume each hydro-fracture event creates an along-ice-flow slippery-patch with a length scale $l$ equal to twice the idealised-blister diameter. This assumption likely produces a maximum estimate of the actual water-lubricated spatial extent, given that the idealised-blister diameter is calculated from a lake's observed maximum volume. Following ref. 15, the perturbed along-flow, tensile ice-sheet surface stress at the inland boundary of the slippery patch can be estimated through $\rho g l \alpha /4$, where $\rho$ is ice density (917 kg m³), $g$ is gravitational acceleration (9.81 m s⁻²), and $\alpha$ is the ice-sheet surface slope over the length of the slippery patch[15]. For the 39 in-cluster hydro-fracture events, the average slippery-patch length $l$ is 8.4 km, and the average ice-sheet surface slope $\alpha$ across these slippery patches is 0.006, yielding a representative stress value at this inland boundary of + 113 kPa. This estimate uses the depth-averaged shallow ice stream approximation and has been validated against simulations that use Stokes flow[15]. We choose to model the viscous stresses during the time period when the slippery patch is present[15]—as compared to previous approaches that use fully relaxed, re-equilibrated timescales[13,14,20]—because the timescale of interest for investigating hydro-fracture events that take place within daily evolving subglacial drainage systems is hours-to-days, not multiple weeks. These viscous stresses will exponentially decay moving inland of the slippery patch[15], and we use the average slippery-patch length for the 39 in-cluster hydro-fracture events, alongside the observational constraints provided by the mechanistic lake-drainage catalogues, to guide our analyses of which additional hydro-fracture and moulin-lake drainage events may fall inside of perturbed-stress regions created by slippery ice-sheet bed patches.

For modelling consideration (3), we estimate the locations of subglacial-flood-propagation fronts produced by hydro-fracture events using GNSS observations of the speed of three different travelling subglacial blisters within our array (Supplementary Information Table S9). Observed propagation speeds range from 0.33–0.45 m s⁻¹; fit within the observed, expected range for ice-sheet supraglacial lake drainages[55]; and, are slightly faster than the ~ 0.3 m s⁻¹ speed reported for a Paakitsoq region event[17]. We settle on a value of 0.4 m s⁻¹ (35 km d⁻¹) for our modelled subglacial-flood-propagation fronts (e.g., Figs. 7b, 8b). We take the potential blister-propagation pathway to encompass both down-ice and down-subglacial-pathway orientations because, for many regions in our study area, subglacial-drainage pathways develop at moderate angle to ice-flow direction due to basal ridges running orthogonal to ice flow[48,49,70] (e.g., Fig. 7a). While we consider all hydro-fracture and moulin-drainage events encompassed within flood-propagation fronts as plausible examples of inter-event triggering, high-pressure subglacial blisters shed water to the surrounding drainage system as they travel[47], causing blisters to lose height[55], which reduces their capacity to induce tensile ice-sheet surface stresses through ice-sheet uplift[12]. Thus, while a velocity response can be observed without corresponding ice-sheet uplift during latter-stage floods[55], these events may be less likely to drive large, tensile ice-surface stresses[12].

## Data availability

Sentinel-1A/B Synthetic Aperture Radar images from the European Space Agency were accessed via the National Snow and Ice Data Centre (NSIDC) MEaSUREs Greenland Image Mosaics from Sentinel-1A and -1B, Version 4 archive[71] (https://nsidc.org/data/nsidc-0723/versions/4). Sentinel-2 Imagery from the European Space Agency were accessed via Google Earth Engine. MEaSUREs InSAR velocity data[65] are archived at the NSIDC (https://nsidc.org/data/nsidc-0478/versions/2). Campaign GNSS and lake pressure-logger data are archived[72] at the GAGE Facility operated by the EarthScope Consortium (https://doi.org/10.7283/HQ02-0044). The mechanistic lake-drainage catalogues for 2022 and 2023 generated during this study are archived on *Zenodo*[73,74] (https://zenodo.org/records/19387943; https://zenodo.org/records/19387985). The data, model output, and code required to recreate all display items presented in this study are available on *Zenodo*[75] (https://zenodo.org/records/19387821) and Figshare[76] (https://doi.org/10.6084/m9.figshare.31212229). All display items presented in the main manuscript and Supplementary Information can be reproduced from data shared in the Figshare and *Zenodo* repositories[73–76].

## Code availability

The code for the FASTER algorithm adapted for this work is described in published literature[30] and archived at the Apollo–University of Cambridge Repository[77] (https://doi.org/10.17863/CAM.25769). The code for the subglacial hydrology model adapted for this work is described in published literature[48,53,55] and archived on *Zenodo*[78] (https://doi.org/10.5281/zenodo.7023662). The code for modelling elastic, ice-sheet stress change following hydro-fracture events, adapted for this work, is described in published literature[12] and archived on *Zenodo*[79] (https://doi.org/10.5281/zenodo.10650188).

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

## Acknowledgements

Logistical and instrumental support was provided by, principally, S. Gossman, A. Rosic, S. Wilson, and J. Stoddard of Polar Field Services and N. Bayou and J. Pettit of EarthScope Consortium. Geospatial support for this work and DEMs were provided by the Polar Geospatial Centre under NSF-OPP award 2003464. We thank Peter Davidsen, Ib Hansen, and Finn Siegstad, of the AirGreenland cargo and charter departments, and Air-Greenland helicopter pilots Geir Bjørnar Akse, Nils Norin, Jonas Olsson, Emil Adsten, and Jørgen Eilertsen, for their contributions to the fieldwork. We thank K. Arnold of Iris Alpine for mountaineering education and on-ice guiding. We thank S. B. Das, M. D. Behn, J. J. McGuire, W. Fan, K. Arnell, E. Towns, N. Lau, and M. Bingham for collaboration in safety training, in-Ilulissat operations, and for early critique in support of this work. Funding: L.A.S. discloses support for the research of this work from the UK Natural Environmental Research Council (NE/Y002369/1), Oxford University Press, and the Radcliffe Institute for Advanced Study at Harvard University. M.N. discloses support for the research of this work from the National Science Foundation's Office of Polar Programs (OPP-2003464). C.-Y.L. discloses support for the research of this work from the National Science Foundation's Office of Polar Programs (OPP-2344690).

## Author contributions

L.A.S. wrote the proposals that funded this work and, in so doing, designed the GNSS array, as mentored by M.N. M.N. and L.A.S. co-led the five field campaigns required to complete this work, as well as their logistical planning. M.O., S.L., J.R. and G.L. participated in these field campaigns. C.-Y.L. supported J.R.'s fieldwork participation through laboratory funds and mentorship. Alongside EarthScope colleagues, M.O. designed, built, and tested the GNSS monumentation, power systems, and their integration with the Xeos Resolute receivers. S.L. processed the GNSS data, as mentored by M.N. E.F. developed the initial strain-rate and bed-separation analysis, as mentored by L.A.S. N.T. developed the initial overspill-connection and moulin-mapping analysis, as mentored by L.A.S. J.R. developed the initial slippery-patch analysis, as mentored by C.-Y.L. L.A.S. performed all other data analysis and modelling and wrote the paper, with input from M.N. S.L., J.R., C.-Y.L. and G.L. commented on the paper.

## Competing interests

The authors declare no competing interests.
