## [Transparent Peer Review file · Nature Communications]

Ice-sheet hydro-fracture not advanced inland by lower-elevation lake drainages in Kalaallit Nunaat

Corresponding Author: Professor Laura Stevens

Version 0:

Reviewer comments:

Reviewer #1

(Remarks to the Author)

In the revised version of 'Ice-sheet hydro-fracture not advanced inland by low-elevation lake drainages,' Stevens et al. more clearly present an in-situ, satellite, and modeling case against pervasive stress perturbation triggered supraglacial lake drainage cascades.

Overall, my previous comments are well addressed, and the combination of observations presented clearly demonstrate that, in the region of study, upstream triggering of cascading lake drainages is unlikely to occur. Based on the likely – though not directly confirmed – similarity between the region studied here and regions in previous studies, pervasive upstream propagating lake drainages are unlikely to cause high elevation surface to bed connections, as previously postulated. This is likely due to the length scale over which the necessary stresses to trigger another lake hydrofracture is very likely shorter than the distances between individual lakes. However, I do not think that the results can completely eliminate the potential of cascading lake drainage as a mechanism, as suggested by the title, because this potential is dictated by ice and bed conditions (which vary over long time scales), which dictate the coupling length scale and inter-lake distance, as well as the state of the lake at the time of hydrofracture.

I only have minor comments, listed below:

No line: There are multiple preprints referenced, in some instances, as the sole reference.

1. It would be worth focusing the title more specifically on Greenland or the region of Greenland 79-81. This sentence reads a bit awkwardly.

82-84. Could 'lower elevation' vs 'higher elevation' be characterized in some way? Relative to the long-term ELA? Being non-specific could lead to misinterpretation.

87. Based on some of the comments in the response to my review and the differences in previous work, I would recommend defining what is meant by 'long-range'. Is it substantially greater than the expected coupling length scale or not?

178-180. The study of Christoffersen et al. is focused on the Kangerlussauq region, while the ROI here is located south of Jakobshavn. Because the locations are different, as are the years of study, I think this statement should be revised to more clearly state that this region does not demonstrate the high number of hydrofracture events modeled in Christoffersen et al.

191. Because it's only a single paper referenced here, consider 'can lead' or 'may lead' instead of 'leads'

211-216. I appreciate the definition of these mechanisms.

257-261. I am going to preface this statement by saying that I am trying to be helpful since as the previous reviewer 1, not all my comments were appreciated. The text justifying the separating these clusters is insufficient. I believe that they could be driven by elevation-dependent lake filling rates; however, there needs to be more referenced or observational/modeling support that 15-20km is too far to see the necessary elevation in tensile stresses. Is there a lack of strain rate perturbation?

272. This references a preprint.

273. 'would' or 'would likely'?

292/293. Not sure that references are warranted here since the phrase is referring to the analysis performed within.

335-341. Could 'small' be quantified and tied to literature?

350-361. It would be worth discussing the reasons for shorter coupling length scales than 'previously assumed' based on the work of Kamb and Echelmeyer (1986) and Kavanaugh and Cuffey (2009). The observations presented here suggest that the coupling length scales in the ROI are more like alpine glaciers, perhaps due to ice surface slopes and velocities. This in combination with lake densities, as previously mentioned in the manuscript, align to inhibit widespread up-elevation lake

drainage cascades.

Reviewer #2

(Remarks to the Author)

Review comment on "Ice-sheet hydro-fracture not advanced inland by low-elevation lake drainages" by Laura A. Stevens et al.

This is my second time reviewing this paper. The revised version includes more figures in the main text, which makes it easier to follow the elaborative work. The scientific merit of this study is clearer than before because the shortcomings of previous studies, i.e. limitations of temporal aliasing based on remote sensing data, are clearly described. I think the manuscript is more or less ready to report the results of a very well-organized study, which will make an important contribution to glacier hydrology and dynamics. I leave only minor comments below.

Title: I still think "Greenland" is missing in the title.

Line 69: "we observe hydro-fracture events" >> we observe ice motion?

Line 89: "Remote observation of hydro-fracture" >> "Remote observation of lake drainage"

Line 131: "FASTER algorithm" >> Please spell out, and refer to a citation and/or the Method section.

Introduction:

The first four paragraphs (Line 32-87) serve as a nice introduction of the paper. Then I expect Results, but the first four paragraphs of the section "Remote observation of hydro-fracture" (90-137) still explain the shortcomings of previous studies. Can you organize this part to make a clear transition from introduction to study results?

Not only this part, the structure and flow of the paper are not smooth to me. The problem is that the methods, results, and discussion are not clearly separated. However, this is largely because of the journal style and difficult for me to suggest restructuring the paper.

Reviewer #3

(Remarks to the Author)

Summary:

This paper uses a 22-station GNSS array spanning 950–1400 m elevation in Central West Greenland to test whether hydrofracture events at lower-elevation supraglacial lakes can trigger hydrofracture at higher elevations. The authors find no evidence for long-range triggering: strain rates across higher-elevation lake basins remain unperturbed during lower-elevation hydrofracture events. Combined with a new mechanistic lake-drainage classification (distinguishing hydrofracture from overspill and moulin drainage), the paper challenges previous claims of cascading hydrofracture across tens of kilometers.

My expertise is in GNSS analysis and interpretation. I focus my review on the geodetic components of this work, which provide the "ground truth" central to the paper's novel contribution. I am not in a position to evaluate the statistical clustering methodology or ice-sheet mechanics modeling in detail.

Overall assessment:

This is a well-executed observational study addressing an important question for ice-sheet dynamics under climate warming. The two-season GNSS deployment represents a significant field effort, and the mechanistic drainage classification is a valuable methodological advance. The central conclusion — that hydrofracture initiation migrates with surface melt rather than being accelerated by distant drainage events — is well-supported by the strain-rate observations presented. I recommend publication following minor revisions to improve reproducibility of the GNSS methodology.

Major comments:

1. GNSS methodology needs additional detail for reproducibility. While the Methods section provides basic processing information (TRACK/GAMIT, kinematic mode, KAGA base station, 5-s sampling, 15-s position output), several standard details are missing:

- o What is the baseline length to KAGA? Long baselines in kinematic mode can degrade vertical precision, which matters for bed-separation estimates.
- o How were tropospheric and ionospheric delays handled? This is important for Greenland where atmospheric conditions can significantly affect solutions.
- o What is the monument design? The authors mention "periodic, multipath noise" requiring 18–36 hr smoothing — a brief description or photo of the installation would help readers understand the noise environment.

2. Strain-rate detection threshold should be quantified. The paper's key claim is that strain rates at higher-elevation basins are "unperturbed" during lower-elevation hydrofracture events. What strain-rate perturbation would be detectable given the measurement precision? Quantifying this would strengthen the argument — the difference between "we saw no signal" and "any signal was below $X \text{ yr}^{-1}$ " is important.

Minor comments:

- L82–83: Can you quantify "unperturbed"? A bound on detectable strain-rate perturbation would strengthen this central claim.
- L94: The 15-s sampling rate is mentioned only in Methods. Given the paper's argument about temporal resolution being critical (vs. 24–144 hr satellite aliasing), stating this in the main text would reinforce the point.
- Figures 5–6: Error bars or uncertainty bands on the strain-rate time series would help readers assess which excursions are significant versus noise.
- Methods (L716–725): The quality-control criteria are clear, but what fraction of data was rejected? This gives readers a sense of data quality.

Recommendation:

Minor revision. The science is sound and the conclusions are well-supported. The requested methodological clarifications are standard for geodetic work and should be straightforward to address.

Version 1:

Reviewer comments:

Reviewer #1

(Remarks to the Author)

Thank you for the revisions to 'Ice-sheet hydro-fracture not advanced inland by lower-elevation lake drainages in Kalaallit Nunaat.' All my concerns have been addressed. Well done on an interesting manuscript.

(Remarks on code availability)

The referenced codes are usable and well documented.

Reviewer #2

(Remarks to the Author)

I think the manuscript has been revised adequately by addressing the reviewers' comments. Some of the arguments and data are more supported by additional texts. Overall, this is a fine piece of work, based on valuable field data, solid analyses, and sophisticated modelling. My only concern is the title. I support the idea of using Greenlandic, but it is still not common in literature to replace "Greenland" with "Kalaallit Nunaat". To help readers, I suggest writing "Kalaallit Nunaat (Greenland)" in the title and the main text (Line 65).

(Remarks on code availability)

Reviewer #3

(Remarks to the Author)

My expertise is in GNSS analysis and interpretation. My comments in the previous review focused on this part of the paper. Reviewing the revised manuscript and the author responses to my comments, I am satisfied that the parts of the paper dealing with the GNSS analysis are ready to be published.

(Remarks on code availability)

Reviewer #1 (Remarks to the Author):

In the revised version of ‘Ice-sheet hydro-fracture not advanced inland by low-elevation lake drainages,’ Stevens et al. more clearly present an in-situ, satellite, and modeling case against pervasive stress perturbation triggered supraglacial lake drainage cascades.

Overall, my previous comments are well addressed, and the combination of observations presented clearly demonstrate that, in the region of study, upstream triggering of cascading lake drainages is unlikely to occur. Based on the likely – though not directly confirmed – similarity between the region studied here and regions in previous studies, pervasive upstream propagating lake drainages are unlikely to cause high elevation surface to bed connections, as previously postulated. This is likely due to the length scale over which the necessary stresses to trigger another lake hydrofracture is very likely shorter than the distances between individual lakes. However, I do not think that the results can completely eliminate the potential of cascading lake drainage as a mechanism, as suggested by the title, because this potential is dictated by ice and bed conditions (which vary over long time scales), which dictate the coupling length scale and inter-lake distance, as well as the state of the lake at the time of hydrofracture. I only have minor comments, listed below.

No line: There are multiple preprints referenced, in some instances, as the sole reference.

Thank you to Reviewer 1 for two sets of reviews! The version of the manuscript read by Reviewer 1 referenced two preprints. One of these preprints (Gjerde et al., 2025) has since appeared in final, peer-reviewed form; the citation has been updated accordingly. We believe the remaining citations to preprint Rines et al. (2025) are appropriate, consistent with current community practice, and align with the Nature Communications’ Editorial Policy that encourages the citation of preprints within articles under consideration for publication (Preprints & Conference Proceedings). We welcome scrutiny of the assumptions, methods, and conclusions of all cited studies in our manuscript; no specific concerns were raised here.

1. It would be worth focusing the title more specifically on Greenland or the region of Greenland

Title revised to “Ice-sheet hydro-fracture not advanced inland by lower-elevation lake drainages in Kalaallit Nunaat”.

79-81. This sentence reads a bit awkwardly.

Sentence revised to “These instances amount to clusters of handfuls of lake-drainage events, whose small sizes comprise 2–7% of the lakes in the study region with volumes viable for full-ice-thickness hydro-fracture at the times that the clusters take place.”

82-84. Could ‘lower elevation’ vs ‘higher elevation’ be characterized in some way? Relative to the long-term ELA? Being non-specific could lead to misinterpretation.

Sentence reads: “Our ground-truth, GNSS observations repeatedly show unperturbed strain rates across higher-elevation ice-sheet regions and lake basins while local clusters of hydro-fracture events transpire at lower elevation.”

The “lower”-versus-“higher” word choice indicates the relative elevation of hydro-fracture events to other lake basins and is not intended to indicate a specific elevation contour. The long-term equilibrium line altitude (ELA) would not be a useful elevation band for describing the relative elevation of hydro-fracture events observed in our study region because all hydro-fracture events observed take place in lake basins located >150 m (in elevation) below the ELA. The long-term equilibrium line altitude (ELA) in the study region sits around 1700 m a.s.l., when estimated using 1-km-resolution RACMO surface mass balance (SMB) products averaged over 1958–2019 available in QGreenland (Revision Figure 1;

Moon et al., 2021; Noël et al., 2019). The observed hydro-fracture events in our study area are located from 700–1300 m a.s.l. in 2022 and from 700–1550 in 2023 (Manuscript Figure 1).

To provide this context to the reader, we have added the contour of the long-term ELA to the revised Manuscript Figure 1, and now give the elevation of the long-term ELA in Methods section “Mechanistic lake-drainage catalogues”, where we previously just described the elevation of ice-slab onset. Revised sentence reads: “The onset of ice slabs at this latitude (68–69° N) occurs at ~1600 m a.s.l. (MacFerrin et al., 2019), and the long-term elevation of the equilibrium line altitude (ELA), calculated using the 1-km-resolution RACMO surface mass balance product averaged from 1958–2019, sits at ~1700 m a.s.l. (Noël et al., 2019).”

To ensure that it is clear to the reader that it is the relative elevation between lake basins that our analysis entails, we have reviewed the manuscript to change all previous usages of “low-elevation lakes” to “lower-elevation lakes” to avoid this potential misinterpretation. The strength of this wording and analysis approach is that, for each hydro-fracture event observed, we are able to look at strain-rates across lake basins at higher elevations (relative to the observed hydro-fracture event taking place at a lower elevation). In this way, we are not limited to comparing hydro-fracture events below versus above a pre-defined elevation contour, but, rather, we can assess the impact of hydro-fracture events taking place at any lower elevation on the ice sheet on our GNSS-observed strain rates across lake basins at 950, 1150, and 1350 m a.s.l. (Manuscript Figures 5, 6). This analysis approach allows us to observe the strain-rate response of higher-elevation lake basins to lower-elevation hydro-fracture events in a couple different instances in each melt season, providing supporting evidence that the strain-rate isolation we observe between lake basins at different elevations largely occurs irrespective of the elevation of the lake basins themselves. This set of observations points towards strain-rate isolation between lake basins at different elevations likely being a coherent characteristic for lakes forming at elevations spanning most of the ablation zone.

87. Based on some of the comments in the response to my review and the differences in previous work, I would recommend defining what is meant by ‘long-range’. Is it substantially greater than the expected coupling length scale or not?

Sentence revised to clarify that by “long-range” we are referring to a tens-of-kilometre distance, aligning with the Christoffersen et al. (2018) statement that hydro-fracture events are related over distances as far apart as 80 km.

The sentence now reads: “This isolation between lower- and higher-elevation basins strongly argues that—although surface lakes advance higher in a warming climate (Fan et al., 2024)—the ability of this meltwater to access the bed and augment ice-flow speeds at high elevations is not accelerated by the process of long-range (i.e., over tens-of-kilometre distances), hydro-fracture-event triggering.”

178-180. The study of Christoffersen et al. is focused on the Kangerlussauq region, while the ROI here is located south of Jakobshavn. Because the locations are different, as are the years of study, I think this statement should be revised to more clearly state that this region does not demonstrate the high number of hydrofracture events modeled in Christoffersen et al.

The sentence reads: “When we analyze lake-drainage timing with the knowledge of drainage mechanism, and accounting appropriately for data gaps, we find no evidence for the >100 hydro-fracture events previously reported to occur only over a few days within a similar ice-sheet region in southwestern Greenland (Christoffersen et al., 2018).”

We elect to keep the “reported to occur” wording as is. Revising the sentence to say that the ROI of our study region “does not demonstrate the high number of hydrofracture events modeled in Christoffersen et al.” (Reviewer 1 comment above) would communicate to the reader that the high number of hydro-fracture events reported to occur, modelled, and mechanistically interpreted by Christoffersen et al. (2018) was a realistic number of hydro-fracture events. The number of hydro-

fracture events reported to occur, modelled, and mechanistically interpreted by Christoffersen et al. (2018) was biased high, yielding over four hydro-fracture events per lake, on average, during the investigated melt season.

Revision Figure 1: A QGreenland screen grab showing: (grey lines) BedMachine v5 (Morlighem et al., 2022) elevation contours with contour intervals plotted every 100 m a.s.l.; (blue lines) elevation contours with contour intervals plotted every 500 m a.s.l.; and (colormap) long-term annual runoff estimated using 1-km-resolution RACMO outputs averaged over 1958–2019 available in QGreenland (Moon et al., 2021; Noël et al., 2019), where the purple start of the color map is the first, non-zero value of runoff (i.e., $SMB < 0$). Dark grey lake outlines show supraglacial-lake maximum extents during 2000–2010 (Yang et al., 2015).

191. Because it's only a single paper referenced here, consider 'can lead' or 'may lead' instead of 'leads'

The sentence reads: "Our analysis demonstrates that lumping together all lake-volume-reduction events irrespective of lake-volume-loss mechanism (e.g., Christoffersen et al. (2018)) yields temporal clustering that dominantly tracks overspill-event timing (Fig. 4e–h), and leads to an incorrect inference of long-range hydro-fracture-event triggering as the mechanism responsible for over three quarters of lake-drainage events (Christoffersen et al., 2018)".

In this sentence, "leads" is most appropriate because Christoffersen et al. (2018) is the only study to lump together all lake-volume-reduction events irrespective of lake-volume-loss "rapid" versus "slow" classification, and is the only study to infer that long-range, hydro-fracture-event triggering is the mechanism responsible for over three quarters of all lake-drainage events. The citation makes clear that these are methodological choices made by that specific study, and not others. The "three quarters of lake-drainage events" proportion is the proportion given by Christoffersen et al. (2018), as described earlier on in our manuscript in the second paragraph of the Introduction.

211-216. I appreciate the definition of these mechanisms.

Thanks.

257-261. I am going to preface this statement by saying that I am trying to be helpful since as the previous reviewer 1, not all my comments were appreciated. The text justifying the separating these clusters is insufficient. I believe that they could be driven by elevation-dependent lake filling rates; however, there needs to be more referenced or observational/modeling support that 15-20km is too far to see the necessary elevation in tensile stresses. Is there a lack of strain rate perturbation?

The sentence reads: "Similarly, we split clusters C2 in 2022 and C5 in 2023 into two spatially independent subclusters because the subclusters occur 15–20 km far apart in the across-ice-flow direction and are not plausibly related through subglacial pathways (Figs. 5a, 6a)."

The hydro-fracture events within the C2 and C5 clusters did not occur within lake basins instrumented by the GNSS array, so we are not able to observe strain-rate perturbations across these in-cluster hydro-fracture events. We can, however, use observations of the bed topography; observations of lake-drainage mechanism of the 12–16 lakes separating the subclusters; and model-derived estimates of (1) subglacial hydrologic routing to identify physically plausible flood propagation routes following the C2 and C5 hydro-fracture events, and (2) the modelled extent of high-tensile-stress regions to assess the across-ice-flow extent of stress perturbations. We now include these details in the main text.

The revised paragraph reads: "By contrast, for some statistically significant temporal clusters, one or two hydro-fracture events are located outside of relevant subglacial pathways and high-tensile-stress regions produced by the other in-cluster events. We consider these hydro-fracture events not to be plausibly related (e.g., lake L2D of cluster C3 in 2022; Fig. 8). Similarly, we split clusters C2 in 2022 and C5 in 2023 into two spatially independent subclusters because the subclusters occur 15–20 km far apart in the across-ice-flow direction, with the region separating the subclusters hosting 12 (2022; Fig. 5a) to 16 (2023; Fig. 6a) lakes that do not drain via hydro-fracture during the time of the clusters. Similarly, the modelled extents of high-tensile-stress regions from the C2 and C5 events show a ~15-km-stretch of no elevated surface stress between the subclusters (Supplementary Information Figs. S2.1, S5.1). Moreover, the C2 and C5 subclusters are unlikely to be related through subglacial pathways, with southern subcluster subglacial floods propagating westward along subglacial lows to beneath MHIH (Fig. 1b,c; Supplementary Information Figs. S2.2c, S5.2c), and northern subclusters subglacial floods likely propagating northwestward to regions north of the GNSS array, or southwestward to beneath the GNSS-instrumented basins at 1150 m a.s.l. (Fig. 1b,c; Supplementary Information Figs. S2.1a, S5.1a). The sizes of these four subclusters lie within expected cluster-size distributions (Fig. 3b,e). That statistically significant temporal clusters contain hydro-fracture events at similar altitudes—yet spaced too far apart to be plausibly related via inter-lake, hydro-fracture-event

triggering—exemplifies the temporal clustering of lake drainage that arises stochastically due to elevation-dependent lake-filling rates (Selmes et al., 2011; Arnold et al., 2014; Stevens et al., 2024). ”

272. This references a preprint.

The sentence reads: “In 2022, lake L2C drains via hydro-fracture within a ~2-hr, GNSS-constrained time window of the L2A and L2B hydro-fracture events (Figs. 5, 8a; Supplementary Information Text S3), yielding a potential example of inter-lake, hydro-fracture event triggering via the slippery-patch mechanism (Rines et al., 2025).”

The sentence states that this set of hydro-fracture events yields a “potential example” of inter-lake, hydro-fracture event triggering via the slippery-patch mechanism detailed in the preprint Rines et al. (2025). We believe this citation is appropriate. The citation is consistent with journal policy.

273. ‘would’ or ‘would likely’?

Revised wording to “would likely”.

292/293. Not sure that references are warranted here since the phrase is referring to the analysis performed within.

References removed.

335-341. Could ‘small’ be quantified and tied to literature?

Yes, we now quantify “small” with regards to (1) our detectable lower bound on strain-rate perturbations of $\pm 0.002 \text{ yr}^{-1}$ added in response to Reviewer 3’s comments, and to (2) another observation of extensional strain-rate transients due to passing subglacial floods reported in the Paakitsoq region by Andrews et al. (2018).

Sentences revised to: “Two “tiepoint” GNSS stations, QIET and MLOW, located within the 1000–1100 m a.s.l. elevation band, allow us to investigate hydro-fracture-induced strain-rate excursions for the ice-sheet region between the 950 and 1150 m a.s.l. GNSS-instrumented basins (Figs. 5a, 6a). Here, hydro-fracture-induced speed-ups at SQ12 result in positive strain-rate excursions between SQ12 and QIET (Figs. 5c, 6c), though the magnitude of these strain-rate excursions are small ($< 0.025 \text{ yr}^{-1}$), aligning with a $+0.015 \text{ yr}^{-1}$ longitudinal strain-rate perturbation observed up to ~4 km inland of three lower-elevation, lake-drainage events in the Paakitsoq region (Andrews et al., 2018). Moreover, none of the five or so overspill-type lakes that form in between SQ12 and QIET drain via hydro-fracture in either year. Rates of extension between the two tiepoint stations and the GNSS stations at 1150 m a.s.l. are unperturbed (i.e., below the measurement-detection threshold of $\pm 0.002 \text{ yr}^{-1}$) as subglacial-flood events transpire in the 850–950 m a.s.l. elevation band (Figs. 5c, 6c).”

350-361. It would be worth discussing the reasons for shorter coupling length scales than ‘previously assumed’ based on the work of Kamb and Echelmeyer (1986) and Kavanaugh and Cuffey (2009). The observations presented here suggest that the coupling length scales in the ROI are more like alpine glaciers, perhaps due to ice surface slopes and velocities. This in combination with lake densities, as previously mentioned in the manuscript, align to inhibit widespread up-elevation lake drainage cascades.

*The reviewer’s comment refers to the following paragraph in the previously submitted manuscript, where we do not use the phrase “previously assumed”. (We do not use this phrase anywhere in the manuscript.) The paragraph formerly read (**emphasis added**): “The absence of a strain-rate response within higher-elevation lake basins during hydro-fracture events in lower-elevation regions argues in support of a small limit, of a few ice thicknesses, on the distance over which lower-elevation ice-flow modulation can promote hydro-fracture initiation at higher elevations. The existence of dozens of*

overspill-type lakes of sufficient volume to hydro-fracture, located immediately inland and down-flow of the hydro-fracture event clusters (Figs. 5a, 6a), empirically confirms the **short length scale** over which inter-lake, hydro-fracture event triggering takes place. Complex spatial patterns of ice-sheet surface stress observed immediately following hydro-fracture events constrained by denser GNSS arrays¹²; basal ridges separating lake-forming regions⁴⁹; and hypothesized narrow subglacial flood widths (<2 km) that follow bed topographic lows^{12,51,52} may all help explain why many candidate lakes viable for hydro-fracture do not, in fact, fracture, and instead drain via overspill, or freeze over.”

We acknowledge that our use of the term “short length scale” in this paragraph is an ambiguous descriptor due to its similarity to the term “coupling length scales”. We want to clarify that the term “coupling length scales,” as used by Kamb and Echelmeyer (1986), differs from our descriptive phrase of “the short length scale over which inter-lake, hydro-fracture event triggering takes place.” In our revised paragraph below, we therefore now adopt the wording of “the short distance over which inter-lake, hydro-fracture event triggering takes place” to avoid this misinterpretation.

We also want to clarify that Kamb and Echelmeyer (1986)’s analysis showed that “theoretically, the coupling length ℓ is generally in the range of one to three times the ice thickness” (Kamb and Echelmeyer, 1986). Although Kamb and Echelmeyer focused on variations in bed slope, which is different from Rines et al. (2024)’s focus on slippery bed patches, the local influence is consistent. More specifically, the coupling length as defined in Kamb and Echelmeyer (1986) and Rines et al. (2024) are both e-folding length scales over which dramatic velocity or stress decay occurs. The coupling length scale does not equate to the “length scale over which inter-lake, hydro-fracture event triggering takes place,” as the latter would require specifying a threshold stress at which hydrofracture-induced lake drainage can be triggered. The magnitude of stress perturbation needed to trigger a hydro-fracture event is likely around +100–200 kPa, meaning that this stress-perturbation will happen while the gradient in flow speeds (i.e., strain rates) inland of the hydro-fracture event is still an appreciable size. This high gradient in flow speeds would occur at a shorter inland distance from the hydro-fracture-event perturbation than the inland distance of the full longitudinal coupling length scale characterized by Kamb and Echelmeyer (1986). Thus, that the observed up-ice-flow distances between hydro-fracture events are shorter than the full longitudinal coupling length scale aligns with the exponential decay in ice-flow speeds observed when moving away from a perturbation (e.g., fast sliding speeds following a hydro-fracture event). That said, it is certainly mechanistically possible that if lakes are very close, a downstream hydro-fracture-triggered lake drainage could locally trigger an upstream lake to drain via hydro-fracture. Here, in the concluding paragraphs of our study, we are simply reporting that there is a lack of direct observations suggesting widespread, hydro-fracture-triggered lake drainage cascades in the inland direction within the region of our observations.

The revised paragraph reads (**emphasis added**): “The absence of a strain-rate response within higher-elevation lake basins during hydro-fracture events in lower-elevation regions argues in support of a small limit, of a few ice thicknesses, on the distance over which lower-elevation ice-flow modulation can promote hydro-fracture initiation at higher elevations. The existence of dozens of overspill-type lakes of sufficient volume to hydro-fracture, located immediately inland and down-flow of the hydro-fracture event clusters (Figs. 5a, 6a), empirically confirms the **short distance** over which inter-lake, hydro-fracture event triggering takes place. Complex spatial patterns of ice-sheet surface stress observed immediately following hydro-fracture events constrained by denser GNSS arrays¹²; basal ridges separating lake-forming regions⁴⁹; and hypothesized narrow subglacial flood widths (<2 km) that follow bed topographic lows^{12,51,52} may all help explain why many candidate lakes viable for hydro-fracture do not, in fact, fracture, and instead drain via overspill, or freeze over.”

Thank you for your review!

Reviewer #2 (Remarks to the Author):

This is my second time reviewing this paper. The revised version includes more figures in the main text, which makes it easier to follow the elaborative work. The scientific merit of this study is clearer than before because the shortcomings of previous studies, i.e. limitations of temporal aliasing based on remote sensing data, are clearly described. I think the manuscript is more or less ready to report the results of a very well-organized study, which will make an important contribution to glacier hydrology and dynamics. I leave only minor comments below.

Thank you to Reviewer 2 for two sets of reviews!

Title: I still think "Greenland" is missing in the title.

Title revised to "Ice-sheet hydro-fracture not advanced inland by lower-elevation lake drainages in Kalaallit Nunaat".

L69: "we observe hydro-fracture events" >> we observe ice motion?

Sentence revised to "With a 22-station Global Navigation Satellite System (GNSS) array installed around seven lake basins in Central West Greenland, we observe ice motion at adequate temporal sampling rates (15-s) to discern inter-lake, hydro-fracture-event triggering potential along a 55-km transect of the ice-sheet ablation zone (Fig. 1)."

L89: "Remote observation of hydro-fracture" >> "Remote observation of lake drainage"

In response to this comment, and the reviewer's comment on the Introduction below, we've now split this section into two different sections to delineate the description of past studies' approaches from the description of our study's results. The section previously titled "Remote observation of hydro-fracture" is now two sections with titles "Shortcomings of apparent-rapidity, lake-drainage classification" and "Feature-based classification of lake-drainage mechanism".

L131: "FASTER algorithm" >> Please spell out, and refer to a citation and/or the Method section.

Sentence revised to define the FASTER acronym and to provide a citation to Williamson et al. (2018).

Revised sentence reads: "Using the Fully Automated Supraglacial lake Tracking at Enhanced Resolution (FASTER) algorithm (Williamson et al., 2018), we track lake surface area for the ~200 lakes that surpass a minimum surface area of 0.165 km² and are located within a 7,189-km², subglacial-catchment-delineated region of interest (ROI) encompassing our GNSS array (Fig. 1; Methods: Mechanistic lake-drainage catalogues)."

Introduction: The first four paragraphs (Line 32-87) serve as a nice introduction of the paper. Then I expect Results, but the first four paragraphs of the section "Remote observation of hydro-fracture" (L90-137) still explain the shortcomings of previous studies. Can you organize this part to make a clear transition from introduction to study results? Not only this part, the structure and flow of the paper are not smooth to me. The problem is that the methods, results, and discussion are not clearly separated. However, this is largely because of the journal style, and it is difficult for me to suggest restructuring the paper.

We have changed the organization of this section of the paper to make a clear transition from the introductory material to our study's results. We've done this by breaking this section of the paper up into two, separate sections with different subheading titles, as detailed in response to line comment L89.

We agree with the reviewer that having most, but not all, of the Methods split off from the rest of the paper means that the structure and flow of the paper are not as smooth as they could be if written in a

different journal style. Since we are doing modelling and analysis work of multiple flavors in this study, we decided to begin each “Results” subsection with enough details on what the work entails before reporting the study’s results. (This approach was also suggested by Reviewer 1 for the modelling work in section “Physical plausibility of event clustering”.) We think that extracting these methodological details from the main text would lower reader comprehension on what analyses we are completing.

Thank you for your review!

Reviewer #3 (Remarks to the Author):

Summary: This paper uses a 22-station GNSS array spanning 950–1400 m elevation in Central West Greenland to test whether hydrofracture events at lower-elevation supraglacial lakes can trigger hydrofracture at higher elevations. The authors find no evidence for long-range triggering: strain rates across higher-elevation lake basins remain unperturbed during lower-elevation hydrofracture events. Combined with a new mechanistic lake-drainage classification (distinguishing hydrofracture from overspill and moulin drainage), the paper challenges previous claims of cascading hydrofracture across tens of kilometers.

My expertise is in GNSS analysis and interpretation. I focus my review on the geodetic components of this work, which provide the "ground truth" central to the paper's novel contribution. I am not in a position to evaluate the statistical clustering methodology or ice-sheet mechanics modeling in detail.

Overall assessment: This is a well-executed observational study addressing an important question for ice-sheet dynamics under climate warming. The two-season GNSS deployment represents a significant field effort, and the mechanistic drainage classification is a valuable methodological advance. The central conclusion — that hydrofracture initiation migrates with surface melt rather than being accelerated by distant drainage events — is well-supported by the strain-rate observations presented. I recommend publication following minor revisions to improve reproducibility of the GNSS methodology.

Major comments:

1. GNSS methodology needs additional detail for reproducibility. While the Methods section provides basic processing information (TRACK/GAMIT, kinematic mode, KAGA base station, 5-s sampling, 15-s position output), several standard details are missing:

→ What is the baseline length to KAGA? Long baselines in kinematic mode can degrade vertical precision, which matters for bed-separation estimates.

→ How were tropospheric and ionospheric delays handled? This is important for Greenland where atmospheric conditions can significantly affect solutions.

→ What is the monument design? The authors mention "periodic, multipath noise" requiring 18–36 hr smoothing — a brief description or photo of the installation would help readers understand the noise environment.

Thank you for your review. We appreciate these suggestions on how to make the GNSS-methodology description more reproducible and quantitative. The Methods section "GNSS data" has been revised to include the following details, which we list here to respond to the reviewer's three sets of questions:

→ The baseline length to KAGA ranges from 50–95 km over the array, moving from the lowest-elevation stations up to the highest-elevation stations. When we designed the array, we included the station QIET as an on-ice station in a region of relatively low ice-flow variability to be used as an on-ice, "fixed" station should the long baselines to KAGA for the higher-elevation stations in our array prove to erode data quality. We ended up not processing the on-ice stations against a "fixed" QIET on-ice base station because poor data quality—majorly related to periodic, multipath noise—was consistently observed across the stations independent of baseline length to KAGA. Revision Figures 2 and 3 below show the consistently poor data quality across the array split up into the relative proportions of data points rejected due to not meeting quality-control criteria of (1) number of allowable TRACK bias flags, (2) being an outlier data point, or (3) having formal errors (one standard deviation) >0.0425 m in the vertical position (Revision Figure 4; Please also see our response to line comment L716–725.)

The reviewer is right to point out that long baselines can degrade vertical precision, and we do observe 1σ error estimates for bed-separation rates to increase slightly with baseline distance (Revision Figures 5 and 6). In our revised manuscript, we now include $\pm 3\sigma$ uncertainty envelopes for all bed-separation

rate timeseries. These timeseries figures are found only in the Supplementary Information. We further provide 1σ error estimates for bed-separation rates for each station, averaged over the melt season days of interest (DOY 165–230), in a new Supplementary Information Table S11. Passing subglacial floods result in bed-separation rates of $>0.3 \text{ m d}^{-1}$, which is far above the $+3\sigma$ lower bound on detectable bed-separation rate perturbations of 0.06 m d^{-1} for our highest-error, inland stations.

→ The tropospheric and ionospheric delays were treated consistently across all stations in the array. The GNSS data were processed as kinematic sites using the ionosphere-free L_C phase observable in the TRACK module of GAMIT/GLOBK. We estimated atmospheric parameters to avoid mapping atmospheric variability into vertical positions, but we did not model ionospheric effects not eliminated by the L_C combination.

→ The monument design was a Septentrio PolaNt-x antenna mounted on an aluminum pole drilled over 3 m into the ice, with the distance from the antenna base to the ice-sheet surface increasing over the ablation season. The ice-sheet surface represents a high-multipath environment, particularly as the melt season progresses and the surface becomes wetter and rough on multiple scales. These details have been added to the Methods section “GNSS data”.

2. Strain-rate detection threshold should be quantified. The paper's key claim is that strain rates at higher-elevation basins are "unperturbed" during lower-elevation hydrofracture events. What strain-rate perturbation would be detectable given the measurement precision? Quantifying this would strengthen the argument — the difference between "we saw no signal" and "any signal was below $X \text{ yr}^{-1}$ " is important.

Longitudinal strain rates between GNSS station pairs are now presented with $\pm 3\sigma$ error envelopes on all figures, and 1σ error estimates for all GNSS station pairs are provided in revised Supplementary Information Table S11. We calculate these error values following Stevens et al. (2024) Equation 2 (i.e., equation of arithmetic error propagation for longitudinal strain rates calculated from GNSS position and velocity estimates). We modify the horizontal-velocity component term in this error-propagation equation to propagate the error in the estimated slope from the sliding, least-squares regressions of station horizontal positions that we use to estimate station horizontal velocities.

Estimated 1σ errors in longitudinal strain rates between GNSS station pairs range from $7 \times 10^{-5} \text{ yr}^{-1}$ to $5 \times 10^{-4} \text{ yr}^{-1}$, when averaged from DOY 165–230 in both melt seasons of the study. With formal errors on TRACK position estimates underestimating the true error in position, and with the data quality for ice-sheet monuments being generally poor (Please see our response to major comment 1 and to line comment L716–725), we take strain-rate perturbations that are three times greater than the high end of our measurement error of $5 \times 10^{-4} \text{ yr}^{-1}$ to be detectable given the measurement precision, placing the bound on a detectable strain-rate perturbation for the dataset at $2 \times 10^{-3} \text{ yr}^{-1}$ (0.002 yr^{-1}). This detectable strain-rate bound aligns with the $\sim 4 \times 10^{-3} \text{ yr}^{-1}$ uncertainties in longitudinal strain rates published in previous work most similar to our study, where authors used on-ice GNSS arrays, with stations spaced a few kilometers apart, in the low-to-mid ablation zone of the Paakitsoq region (Andrews et al., 2018; Poinar and Andrews, 2021).

We have revised relevant sentences in the Introduction; Methods: GNSS-derived quantities; Results: Hydro-fracture across the ablation zone; Figure 5; Figure 6; and many figures in the Supplementary Information to include these details on measurement precision, and to communicate that our deduction that strain rates across higher-elevation basins are unperturbed means that “We saw no change in signal $>0.002 \text{ yr}^{-1}$ during times of lower-elevation hydro-fracture events”. Errors in longitudinal strain rates between GNSS station pairs are now presented with $\pm 3\sigma$ uncertainty envelopes on all figures.

Minor comments:

L82–83: Can you quantify "unperturbed"? A bound on detectable strain-rate perturbation would strengthen this central claim.

Yes, we now quantify our use of “unperturbed” strain-rate modulations using the $>0.002 \text{ yr}^{-1}$ bound on detectable strain-rate perturbations calculated in response to the reviewer’s Major Comment #2. Sentence on L82 now reads: “Our ground-truth, GNSS observations repeatedly show no strain-rate perturbations above our detectable threshold (0.002 yr^{-1}) across higher-elevation ice-sheet regions and lake basins while local clusters of hydro-fracture events transpire at lower elevation.”

We additionally revised two sentences in Results section “Hydro-fracture across the ablation zone” to quantify what our statements of “unperturbed” strain-rate fluctuations mean:

On L328: “During this period, ice-sheet velocities bracketing the sets of lakes increase largely in tandem, resulting in negligible longitudinal strain rates across the basins (Figs. 5b–e, 6b–e), where perturbations in longitudinal strain rates between GNSS station pairs $>0.002 \text{ yr}^{-1}$ are detectable given GNSS measurement precision (Methods: GNSS-derived quantities).”

On L363: “Rates of extension between the two tiepoint stations and the GNSS stations at 1150 m a.s.l. are unperturbed (i.e., below the measurement-detection threshold of $\pm 0.002 \text{ yr}^{-1}$) as subglacial-flood events transpire in the 850–950 m a.s.l. elevation band (Figs. 5c, 6c).”

L94: The 15-s sampling rate is mentioned only in Methods. Given the paper's argument about temporal resolution being critical (vs. 24–144 hr satellite aliasing), stating this in the main text would reinforce the point.

Sentence revised on L72 to highlight the 15-s sampling rate of the GNSS observations: “With a 22-station Global Navigation Satellite System (GNSS) array installed around seven lake basins, we observe ice motion at adequate temporal sampling rates (15-s) to discern inter-lake, hydro-fracture-event triggering potential along a 55-km transect of the Greenland Ice Sheet ablation zone (Fig. 1).”

Figures 5–6: Error bars or uncertainty bands on the strain-rate time series would help readers assess which excursions are significant versus noise.

Uncertainty envelopes of $\pm 3\sigma$ are now plotted on all strain-rate time series in Figures 5 and 6, as well as on all strain-rate time series in the Supplementary Information. Supplementary Table S10 has been revised to report estimated 1σ errors in longitudinal strain rates between GNSS station pairs.

L716–725: The quality-control criteria are clear, but what fraction of data was rejected? This gives readers a sense of data quality.

Methods section updated to communicate that, although the fraction of data rejected generally did not vary from station to station across the array (Revision Figures 2 and 3), the fraction of data rejected to remove periodic, multipath noise in the horizontal and vertical position estimates was high in both years. Points rejected due to having more than two bias flags marked during the TRACK processing amounted to 4% (2022) and 2% (2023) of data points. Points rejected due outlier identification, where points are deemed outliers if they are $>3\sigma$ from the mean value calculated over a sliding, 6-hr window centered on the data point, amounted to 3% (2022) and 3% (2023). Finally, the largest number of points rejected comes from the rejection threshold for formal errors (one standard deviation) $>0.0425 \text{ m}$ in the vertical position; points rejected due to this criterion amounted to 24% (2022) and 30% (2023) of data points. Taken together, these quality-control criteria reject 31% (2022) and 35% (2023) of the data.

These data-acceptance criteria do not affect the main results or interpretations of whether or not strain-rate changes occur at upper-elevation lake basins when lower-elevation lakes drain. The data-acceptance criteria do successfully avoid presenting spurious diurnal signals (i.e., diurnal signals sourced primarily from multipath) to the reader, which could then be misinterpreted by the reader as actual diurnal signals in strain rates sourced from diurnal changes in, for example, surface meltwater

entering moulins (e.g., Andrews et al., 2014). These data-rejection fraction details, and the reasoning motivating these choices, have been added to the Methods section “GNSS data”.

Recommendation: Minor revision. The science is sound and the conclusions are well-supported. The requested methodological clarifications are standard for geodetic work and should be straightforward to address.

Thank you for your review!

Revision Figure 2. Proportion of data points rejected due to the three quality-control criteria for each station during the 2022 melt season (2022/170–230). N.B. The antennae of station SQ13 toppled over during the 2022 melt season; the calculation of data points rejected for this station is only performed from 2022/170–195.

Revision Figure 3. Proportion of data points rejected due to the three quality-control criteria for each station during the 2023 melt season (2023/170–230).

Revision Figure 4. Formal, 1σ errors for TRACK East, North, and Up position estimates for station SQ21 from 2022/190–200, showing periodic fluctuations in error magnitude interpreted to be sourced primarily from multipath. The purple, horizontal line shows the 0.0425-m threshold level for 1σ errors in the Up position estimate, above which the data point is rejected.

Revision Figure 5. Estimated 1σ errors for bed-separation rate \dot{c} estimates for each station during the 2022 melt season (2022/165–230). Values plotted are $\delta\dot{c}$ estimates averaged from 2022/165–230 (i.e., the melt-season window analysed in the study).

Revision Figure 6. Estimated 1σ errors for bed-separation rate \dot{c} estimates for each station during the 2023 melt season (2023/165–230). Values plotted are $\delta\dot{c}$ estimates averaged from 2023/165–230 (i.e., the melt-season window analysed in the study).

References

- Andrews, L. C. et al. (2018), Direct observations of evolving subglacial drainage beneath the Greenland Ice Sheet. *Nature*, 514, 80–83.
- Andrews, L. C. et al. (2018), Seasonal Evolution of the Subglacial Hydrologic System Modified by Supraglacial Lake Drainage in Western Greenland. *JGR Earth Surface* 123, 1479–1496.
- Fan, Y. et al. (2024), Expansion of supraglacial lake area, volume and extent on the Greenland Ice Sheet from 1985 to 2023. *J. Glaciol.* 1–44.
- Gjerde, G., et al. (2025), Seasonal drainage-system evolution beneath the Greenland Ice Sheet inferred from transient speed-up events, *The Cryosphere*, 19, 6149–6169, <https://doi.org/10.5194/tc-19-6149-2025>.
- Kamb, B. & Echelmeyer, K. A. (1986), Stress-Gradient Coupling in Glacier Flow: I. Longitudinal Averaging of the Influence of Ice Thickness and Surface Slope. *J. Glaciol.* 32, 267–284.
- Kavanaugh, J. L., & Cuffey, K. M. (2009). Dynamics and mass balance of Taylor Glacier, Antarctica: 2. Force balance and longitudinal coupling. *JGR Earth Surface*, 114(F4).
- MacFerrin, M. et al. (2019), Rapid expansion of Greenland’s low-permeability ice slabs. *Nature*, 573, 403–407.
- Moon, T., Fisher, M., Harden, L., and T. Stafford (2021). QGreenland (v3.0.0) [software], National Snow and Ice Data Center Available from <https://qgreenland.org/>.
- Morlighem, M. et al. (2022). IceBridge BedMachine Greenland, Version 5 [Data Set]. Boulder, Colorado USA. NASA National Snow and Ice Data Center Distributed Active Archive Center. <https://doi.org/10.5067/GMEVBWFLWA7X>. Date Accessed 06-08-2023.
- Noël, B., W. J. van de Berg, S. Lhermitte, and M. R. van den Broeke (2019), Rapid ablation zone expansion amplifies north Greenland mass loss, *Science Advances*, 5(9), eaaw0123.
- Poinar, K. & Andrews, L. C. (2021), Challenges in predicting Greenland supraglacial lake drainages at the regional scale. *The Cryosphere* 15, 1455–1483.
- Rines, J. H., Lai, C.-Y. & Wang, Y. (2024) Gravity-driven viscous flow over partially lubricated bed. *Preprint* at <https://doi.org/10.48550/ARXIV.2407.20565>.
- Williamson, A. G., Banwell, A. F., Willis, I. C. & Arnold, N. S. (2018), Dual-satellite (Sentinel-2 and Landsat 8) remote sensing of supraglacial lakes in Greenland. *The Cryosphere* 12, 3045–3065.

Reviewer #1 (Remarks to the Author): Thank you for the revisions to 'Ice-sheet hydro-fracture not advanced inland by lower-elevation lake drainages in Kalaallit Nunaat.' All my concerns have been addressed. Well done on an interesting manuscript.

Reviewer #1 (Remarks on code availability): The referenced codes are usable and well documented.

Thank you to Reviewer 1 for multiple sets of reviews, and for reviewing the manuscript's associated code and data repositories.

Reviewer #2 (Remarks to the Author): I think the manuscript has been revised adequately by addressing the reviewers' comments. Some of the arguments and data are more supported by additional texts. Overall, this is a fine piece of work, based on valuable field data, solid analyses, and sophisticated modelling. My only concern is the title. I support the idea of using Greenlandic, but it is still not common in literature to replace "Greenland" with "Kalaallit Nunaat". To help readers, I suggest writing "Kalaallit Nunaat (Greenland)" in the title and the main text (Line 65).

Thank you to Reviewer 2 for multiple sets of reviews. The reviewer's concern about readers needing "help" is unwarranted. First, we must ask, "Which readers does the reviewer think will need help?" Readers who do not know where Kalaallit Nunaat is can learn by searching the place name, reading the paper, or by viewing the maps of Figure 1. The reviewer's line of reasoning ("it is still not common in literature to replace 'Greenland' with 'Kalaallit Nunaat'") employs a postponement argument, which works to defer change until some future, unknown time when said change has already come about. A further fallacy in the reviewer's argument is that people writing papers describing ice-sheet processes that take place in Kalaallit Nunaat determine what is "common in [scientific] literature" that describes ice-sheet processes in Kalaallit Nunaat.

Reviewer #3 (Remarks to the Author): My expertise is in GNSS analysis and interpretation. My comments in the previous review focused on this part of the paper. Reviewing the revised manuscript and the author responses to my comments, I am satisfied that the parts of the paper dealing with the GNSS analysis are ready to be published.

Thank you to Reviewer 3 for reviewing the GNSS-analysis and interpretation components of the work.